# Practical Considerations for Translating Mesenchymal Stromal Cell-Derived Extracellular Vesicles from Bench to Bed

**DOI:** 10.3390/pharmaceutics14081684

**Published:** 2022-08-12

**Authors:** Pauline Po Yee Lui, Yung Tim Leung

**Affiliations:** 1Center for Neuromusculoskeletal Restorative Medicine, Hong Kong SAR, China; 2Department of Orthopaedics and Traumatology, The Chinese University of Hong Kong, Hong Kong SAR, China

**Keywords:** extracellular vesicles, mesenchymal stromal cells, good manufacturing practice, quality control, tendon and ligament injuries

## Abstract

Extracellular vesicles (EVs) derived from mesenchymal stromal cells (MSCs) have shown potential for the treatment of tendon and ligament injuries. This approach can eliminate the need to transplant live cells to the human body, thereby reducing issues related to the maintenance of cell viability and stability and potential erroneous differentiation of transplanted cells to bone or tumor. Despite these advantages, there are practical issues that need to be considered for successful clinical application of MSC-EV-based products in the treatment of tendon and ligament injuries. This review aims to discuss the general and tissue-specific considerations for manufacturing MSC-EVs for clinical translation. Specifically, we will discuss Good Manufacturing Practice (GMP)-compliant manufacturing and quality control (parent cell source, culture conditions, concentration method, quantity, identity, purity and impurities, sterility, potency, reproducibility, storage and formulation), as well as safety and efficacy issues. Special considerations for applying MSC-EVs, such as their compatibility with arthroscopy for the treatment of tendon and ligament injuries, are also highlighted.

## 1. Introduction

### 1.1. MSCs and MSC-EVs for Tissue Repair and Disease Treatment

Mesenchymal stromal cells (MSCs) are pluripotent, non-hematopoietic stem cells with self-renewal capability. They have immunomodulatory, pro-angiogenic and growth promoting effects and are capable of homing to injured sites. Consequently, there has been an extensive interest in their therapeutic use for tissue repair due to conditions such as neurological disorders [1], liver diseases [2], cardiovascular diseases [3], immune-mediated disorders [4], bone fractures [5] and osteoarthritis [6].

Mounting evidence suggests that MSCs enhanced tissue repair via paracrine factors rather than by direct differentiation [7]. As a consequence, researchers are primarily focusing on MSC secretome, which include soluble proteins, lipids and extracellular vesicles (EVs). EVs are membranous, nano-sized vesicles secreted by all cells. They participate in intercellular communication and possess intrinsic therapeutic activity via addressing their vesicular content of DNA, RNA, proteins and other cellular components from a producer cell to a recipient cell. They can be classified into different types, i.e., exosomes (50–150 nm in diameter), microvesicles (100–1000 nm in diameter) and apoptotic bodies (1–5 μm in diameter), depending on their size and biogenesis. EVs have been shown to modulate the immune response [8,9], potentiate tissue regeneration [10] and function as a potential alternative to stem cell-based therapies [11]. Indeed, MSC-EV-enriched preparations have been shown to be as therapeutically effective as their parent cells in different pre-clinical models in head-to-head comparisons [12,13,14]. Genetically modified EVs can also act as delivery vehicles for therapeutic agents [15]. The capacity of EVs has hence spurred interest in their use both as a delivery system and as a drug for the treatment of various disorders.

### 1.2. Advantages of MSC-EV-Based Therapeutics

EV-based therapies in tissue repair and regeneration offer numerous advantages, i.e., lower cost, easier storage and improved safety and patient compliance. First, EVs lack self-replicating ability, thereby reducing the risk of malignant transformation and ectopic tissue formation. Second, while MSCs generally have low immunogenicity and are suitable for allogeneic transplantation [16,17,18,19,20,21,22], the lack of immunogenic cell surface proteins in exosomes likely further reduces immunogenic response after allogeneic transplantation and allows them to be used as an off-the-shelf product. Third, the lipid bilayer membrane also enhances the stability and reduces the toxicity of the EV cargo during in vivo delivery. Unlike soluble proteins in the MSC secretome, it is easier to concentrate EVs, though there may be a loss of some potency of the secretome. Moreover, EVs have the intrinsic ability to cross the epithelium and blood-brain barrier, and hence, may be useful for the delivery of drugs with low oral bioavailability and poor epithelial penetration [23,24,25,26]. In addition, the ability to engineer EVs to improve their specific cell targeting capacity or potentiate their therapeutic properties also makes them good drug carriers and therapeutics for tissue repair. EVs derived from MSCs therefore have the potential to serve as an alternative to MSCs in tissue repair. Early clinical trials investigating MSC-EV-based therapeutics have already begun [27,28,29]. Table 1 summarizes completed and ongoing clinical trials of MSC-EVs registered at www.clinicaltrials.gov (accessed on 30 June 2022).

## 2. MSC- and MSC-EV-Based Therapies for Tendon and Ligament Repair

Tendons and ligaments are subject to high tensile loads and are easily torn as a result of overuse or trauma, resulting in significant pain and disability. Together, tendon and ligament injuries account for 30% of all musculoskeletal consultations [30]. More than 32 million acute and chronic tendon and ligament injuries occur annually in the United States [31]. The outcomes of both conservative treatments and surgical repair of tendon are not satisfactory due to the long healing time, scar tissue formation, adhesion, occasional bone formation and high re-rupture rate. Similar to other tissues, the administration of MSCs is a promising approach for tendon, ligament and tendon-to-bone junction repair [32,33,34]. We have shown that the transplantation of tendon-derived stem cells (TDSCs) promotes tendon regeneration and graft healing after anterior cruciate ligament reconstruction (ACLR) [35,36,37]. Recently, there has been considerable interest in studying the effect of MSC-EVs on tendon and ligament repair, with encouraging results (Table 2) [38,39,40,41,42,43,44,45,46,47,48,49,50,51,52,53,54,55,56,57,58,59,60,61]. However, to date, no clinical trials have been undertaken. As such, the effects of MSC-EVs need to be evaluated in a clinical setting.

The administration of MSC-EVs was generally found to promote cell proliferation and migration [43,44,47,48,49,53,54,55,58,59], suppress tissue inflammation and apoptosis [40,44,46,48,52,55,58,59], modulate inflammatory response of macrophages [38,39,46,48,52,55], increase collagen deposition [43,45,49], reduce fatty infiltration [50] and promote angiogenesis [52] during tendon and ligament repair. The active molecules in MSC-EVs that contribute to tendon and ligament repair are not entirely known. However, numerous studies have shown that MSC-EVs are rich in miRNA, which contribute to tendon and ligament repair. miRNA sequencing showed that human umbilical cord MSC-derived exosomes (HUMSC-Exos) expressed an antagonist to miR-21a-3p, as miR-21a-3p was under expressed in the exosomes compared to the parent cells. The inhibition of exosomal miR-21a-3p in HUMSC-Exos inactivated RelA/p65 (a core element in the NF-kB pathway involved in inflammation and fibrosis) and reduced the protein expression of Cox2 and α-SMA in rat fibroblasts [40]. Additionally, endoplasmic reticulum stress (ERS)-associated proteins (GRP78, CHOP) and pro-apoptotic protein (Bax) were presented in EVs derived from HUMSCs, potentially explaining their anti-adhesive effect on traumatic tendon injury [41]. In addition, HUMSC-Exos we have been shown to promote tendon healing via miR-27b-3p-mediated suppression of ARHGAP5, resulting in RhoA activation and increased proliferation and migration of primary injured tenocytes [42]. In another study, rat EVs derived from bone marrow-derived stromal cells (BMSC-EVs) were reported to express pro-collagen1A2 and MMP14 proteins, which are important factors for tendon extracellular matrix remodeling. Pro-collagen1A2 was expressed on the membrane surface of BMSC-EVs. Pretreatment of BMSC-EVs with trypsin abrogated their effects on tendon cell proliferation and migration and the expression of collagen type I, suggesting that the biological effects of EVs depended on the interaction of membrane-bound proteins with the recipient tendon cells [43]. Furthermore, miRNA sequencing indicated a significantly higher level of miR-29a-3p in HUMSC-Exos compared to HUMSCs [45]. The level of miR-29a-3p in HUMSC-Exos-treated Achilles tendons was also significantly elevated, and HUMSC-Exos overexpressing miR-29a-3p was found to amplify the effects of HUMSC-Exos on tendon healing in vivo [45]. In addition, exosomes derived from TDSCs (TDSC-Exos) have been found to contain miR-144-3p and enhance tendon repair through miR-144-3p-regulated tenocyte proliferation and migration [49]. Finally, exosomes derived from Scx overexpressing PDGFRα(+) BMSCs reduced osteoclastogenesis and improved tendon–bone healing strength via exosomal miR-6924-5p [56].

## 3. Translational Gap and Aim of Review

Unlike single active substances, MSCs and their EVs are comprised of pleiotropic bioactive ingredients, representing a major obstacle for the manufacture of reproducible drug products with consistently high efficacy and safety. The content of EVs is affected by the heterogenicity of the donor MSCs. The phenotypes and biological activities of MSCs depend on their origin (biological niche) or the conditions of potential donors (age, diseases, obesity or unknown factors) [62]. The artificial microenvironment of MSCs such as O_2_ tension, substrate and extracellular matrix cues, culture media, inflammatory stimuli and genetic manipulations can also influence the resulting phenotypes and, hence, paracrine activities [62]. HUMSC-Exos cultured in two different cell culture media showed different surface compositions and cytokine contents [63]. Furthermore, the passage number of MSCs influences the biological activities of EVs. Exosomes isolated from early passage of rat BMSCs exhibited higher neuroprotective potential compared to exosomes derived from later passages [64]. Moreover, the mechanisms of the biogenesis of EVs also influence their content. At least three major subpopulations of EVs produced by different mechanisms, namely exosomes, microvesicles and apoptotic bodies, have been reported. Immortalized E1-MYC 16.3 human embryonic stem cell-derived MSCs were reported to produce at least three distinct 100 nm EV types by different biogenesis pathways that could be distinguished by their membrane lipid composition, proteome and RNA cargos [65]. In one study, BMSC-EVs were fractionated into different density fractions [66]. The different EV gradient fractions were heterogeneous in terms of the quantity and expression of classical exosomal markers. The miRNA and protein profiles of these EV fractions were different, and they also showed differential effects on renal tubular cells in terms of degree of internationalization, stimulation of cell proliferation and inhibition of apoptosis. Therefore, MSC-EVs are heterogeneous, with specific signatures accounting for the biological activity of different subpopulations. EVs produced by a given population of cells cultured under identical conditions are not identical.

Gene therapy medicinal products, somatic cell therapy medicinal products and tissue-engineered products are regulated as advanced therapy products (ATPs) in the authors’ homeland [67] and other places, including Europe (Regulation EC No 1394/2007) [68]. Native cell-free EVs without transgene products are not categorized as ATPs and fall within a regulatory gap. However, EVs are categorized as pharmaceutical products/biological medicinal products, and hence, require clinical trials as investigational new drugs (INDs) before getting marketing approval. To receive an IND, investigators have to demonstrate that their EV-based products fulfill the requirements regarding quality, reproducibility, safety and efficacy. This is especially a challenge for EV-based products due to the heterogeneity and complexity of their composition. While fractionation or sorting may help to identify more homogeneous subpopulations of EVs, careful control of the donor cell source, the cell culture conditions and the EV enrichment process are crucial for clinical translation of EV-based therapies, as they would standardize the EV composition and content.

In this review, we will discuss the general and tendon/ligament-specific considerations of manufacturing MSC-EV-based products for clinical translation. Specially, we will discuss Good Manufacturing Practice (GMP)-compliant manufacturing and quality control (QC), safety and efficacy issues. Special considerations for applying MSC-EVs for the treatment of tendon and ligament injuries are also discussed.

## 4. Considerations for the Manufacturing, Quality Control, Safety and Efficacy of MSC-EVs

The process is the product. A scalable, reproducible and GMP-compliant manufacturing protocol should be followed to produce MSC-EV therapeutics for the promotion of tendon and ligament healing. Standard operating procedures (SOPs) for MSC-EV manufacturing should be followed and the production process should be documented to ensure high batch-to-batch consistency. Newly produced batches should be compared to previous batches regarding physiochemical properties and biological activity.

Besides controlling the manufacturing process, it is important to specify the release criteria of the physio-chemical-biological properties of MSC-EVs for clinical trials or applications. This is vital for ensuring the efficacy, safety and consistency of MSC-EVs as pharmaceutical products. The release criteria are highly product-specific and depend on the cell source and clinical application. Each MSC-EV preparation for a clinical application has its own specifications. The requirements to examine the physiochemical characteristics and sterility are common to different EV products. However, the requirements for testing the identity, purity, impurities, potency, reproducibility, storage and formulation vary with different MSC-EV preparations. The issues that need to be considered for the manufacturing, QC, safety and efficacy of MSC-EVs are discussed below.

### 4.1. MSC-EV Manufacturing

#### 4.1.1. Parent Cell Source

The following factors should be considered for the isolation of EV-producing MSCs for therapeutic applications [69].

Tissue source of MSCsAge, medication, and medical history of donorAllogeneic versus autologous sourceAny priming of MSCsAny genetic modification of MSCsMSC culture conditions (MSc isolation procedures, seeding density, culture volume, culture vessel, oxygen level, culture medium, culture time, cell viability, passaging)MSC storage and recovery conditions

The source of MSCs for EV production determines the manufacturing and QC strategy of the process and the final product. The primary factors to consider include the tissue origin of the cells, the age and medical history of the donor, whether the cells are autologous or allogeneic and whether the cells have been primed or genetically modified. The plasticity of MSCs isolated from different tissues varies, affecting the properties of the isolated EVs [70,71,72]. The proliferation capacity and yield of MSCs also vary based on the tissue of origin [73,74]. Both proliferation capacity and MSCs yield are important factors for large-scale and affordable production of MSCs and EVs for therapeutic applications. The use of cells isolated from tissue similar to the site of transplantation may have an advantage due to the priming of the cells by the local tissue microenvironment. For instance, TDSCs were reported to show higher proliferation, colony-forming ability and multi-lineage differentiation potential compared to paired BMSCs [73] and to form tendon-like tissue after subcutaneous transplantation to a nude mouse model [75]. MSCs isolated from young and healthy subjects are preferred, as aged MSCs and MSCs derived from diabetic or obese patients have been reported to show reduced activities [76,77,78,79,80,81,82]. Clinical examinations are needed to check for signs or symptoms of communicable diseases among donors. In addition, serological tests of at least the human immunodeficiency virus (types 1 and 2), hepatitis virus (B and C) and Treponema pallidum should be performed [83]. The inclusion and exclusion criteria for MSC donors need to be carefully defined and followed.

MSCs are generally reported to have low immunogenicity, and the use of MSC-EVs further reduces safety concerns. As EVs cannot replicate, the safety concerns of using EVs derived from autologous and allogeneic cells for tissue repair are similar. The use of allogeneic cells allows for the large-scale production of EV-based products as off-the-shelf materials and is, hence, preferred. However, the ease, feasibility and ethical concern of getting allogeneic MSCs from patients not suffering from injuries vary with the tissue of origin of MSCs. The identification of a practical MSC source with high proliferative and multi-lineage differential potential is crucial for large-scale manufacturing of MSC-EVs for clinical application. To use allogeneic MSCs for EV production, a two-tier cell banking system, consisting of a fully characterized master cell bank (MCB) and partially characterized working cell banks (WCB), should be set up [84]. A four-tier cell banking system has also been suggested [85]. However, at the early stages, as a first-in man clinical trial, an approach based on only a well-characterized MCB is acceptable (may add a Post Production Cell Bank (PPCB) as discussed below). The criteria for releasing MSCs for EV production should be clearly defined.

#### 4.1.2. Culture Conditions

In additional to the MSC source, the culture conditions should also be considered. How the cells are manipulated (i.e., any stimulation or pre-treatment), cultured and stored, as well as the experimental conditions (i.e., culture medium composition and lot numbers, passage number, days in culture, seeding density, density/confluence at harvest, culture volume, culture containers, surface coatings, oxygen or other gas tensions and frequency and intervals of harvest) can affect the recovery of MSC-EVs, and, hence, should be documented and remain consistent [86].

For the culture of EV-producing MSCs, a GMP-grade culture system should be considered. Both static systems, like flasks, and dynamic systems, such as bioreactors, can be used for MSC culture. For static systems, both standard tissue culture flasks and special-coated flasks such as CellBIND^®^ flasks have been used to produce GMP-compliant EVs [87]. For large-scale production of MSCs, bioreactors are preferred. The metabolic (glucose and lactate) and physical condition (temperature, pH, oxygen, carbon dioxide) readouts during culturing in bioreactors allows for better control and monitoring of the activities of MSCs, which is beneficial for meeting the regulatory requirements for clinical-grade production. Cell culture conditions such as pH, temperature, pO_2_, cell culture duration and metabolic activity (such as glucose and lactose), when applicable, should be defined, recorded and compared to the QC acceptance criteria at the manufacturing phase. Regular cell counting and checking of cell viability should be done. The growth rate of MSCs should be monitored by population doubling time and level.

The use of animal origin-free dissociation enzymes and a culture medium of defined composition is preferred for MSC culture and EV enrichment. The use of a xeno-free medium for cell culture has been reported to reduce the doubling time compared to that of cells cultured in research-grade conditions, to increase exosome yield and to remove 97% of the contaminating proteins [88]. EV-depleted medium should be used for EV isolation. The collection of EVs under ideal, serum-free conditions reduces contamination by non-EV proteins and co-isolation of exogenous EVs from serum. However, starvation may affect the physiology of the EV-producing MSCs, and, hence, EV quantity and quality. Human platelet lysate (HPL)-based EV-depleted medium was shown to be suitable for GMP-compliant MSC propagation and MSC-EV enrichment with retained characteristic of MSC surface marker expression, cell morphology, viability and in vitro differentiation potential [89]. In such cases, we recommend including a non-conditioned medium control to assess the contribution of the medium itself to EV production. If priming agents are added to MSCs to improve the quantity and quality of EV production, the concentration of residual priming molecules in the medium after washing and EV yield per cell equivalent (CE) should be documented and compared to the acceptance criteria. The percentage of dead cells at the time of MSC-EV harvest should be indicated, since even a small percentage of dead cells can release apoptotic bodies exceeding the amount of EVs [86].

A cell passage limit for harvesting EVs should be set, as MSCs show cellular senescence and reduced activity during sub-culturing [90]. To test the stability of MSCs during in vitro passaging, a PPCB, consisting of cells at the limit of in vitro passaging for EV production, can be included.

The cryopreservation and thawing methods, as well as the maximum cryopreservation time, should be validated and defined in the manufacturing process. The cell identity, viability and growth rate should be checked after recovery of MSCs from freezing. According to the Food and Drug Administration (FDA) and European Medicines Agency (EMA) guidelines, the minimum acceptance criteria for the viability of MSC-based products and after cell thawing are ≥80% and ≥70%, respectively [91,92].

The identity of MSC culture needs to be checked. The International Society for Cellular Therapy has published a set of minimal criteria for defining MSCs (plastic-adherent, expression of phenotypic markers and ability to differentiate) [93]. In reality, it is well-known that the ISCT criteria are rarely clinically useful, as the markers are not specific to MSCs [94]. For MSCs isolated from specific tissue, additional tissue-specific markers may be included to confirm the cell source. For instance, TDSCs are known to express high levels of scleraxis (Scx) and tendomodulin (Tnmd), which could be added to the marker list for cell identification [94]. In addition, positive expression of major histocompatibility complex (MHC) class I and low/negative expression of MHC II molecules are frequently evaluated due to their roles in immune cell recognition and, thus, their relationship with potential immune response after allogeneic MSC transplantation.

A potency test mimicking the in vivo effects of MSCs is recommended. Mixed lymphocyte reaction (MLR) and similar approaches (peripheral blood mononuclear cell, PBMC stimulation assay) are commonly used by most groups studying MSCs [95]. However, there are concerns about their robustness as potency release tests. Several variations of the standard assays have been proposed to increase robustness, such as measuring MSC-stimulated induction of regulatory T cells (Treg) and assessing markers indicating the immunosuppressive properties of MSCs, including CD200, TNF-αR, IDO and PD-L1, after IFN-γ stimulation [96,97,98,99,100]. Further validation of these tests is needed.

The manufacture and QC of EVs derived from unmodified cells are simpler compared to those for EVs isolated from apparently more complicated, genetically modified cells. More stringent regulations by health authorities are anticipated. The potential risks due to the vector and the specific transgene used for cell transformation should be identified and controlled in the manufacturing and QC procedures. Furthermore, endotoxin, bacterial, fungal, mycoplasma, adventitious viral, genomic stability (such as karyotyping) and in vitro tumorigenicity tests (such as soft agar colony formation assay) of MSC culture should be done.

In summary, throughout the in vitro expansion process, parent MSCs should be characterized at different checkpoints. The QC panel, including cell culture conditions, cryopreservation and thawing procedures, identity, viability, growth rate, purity, impurities, sterility, stability, safety, and potency of parent MSC banking, is shown in Table 3. MSCs are released for EV isolation only when all the QC results comply with the specifications. As with EV preparations, some MSC samples should be retained for analytical (reference samples) and identification purposes (final product).

#### 4.1.3. Enrichment Method

There is no single optimal method for the concentration or enrichment of MSC-EVs. Different enrichment methods produce different populations [72]. Enrichment methods differ in terms of cost, ease of use, throughput, requirements in time and instrumentation, vesicle loss and the purity of the final product. The MISEV2018 guideline suggests arranging each method on a “recovery vs. specificity” grid [86]. By choosing a highly scalable and automatable concentration method as early as in the pre-clinical stage for MSC-EV isolation, changes in the EV specifications due to the scaling up of the isolation method for clinical trials can be avoided.

Ultracentrifugation (UC) is the most commonly used method for EV enrichment. However, this approach can co-isolate non-EV components, especially for biological fluids such as serum and urine. Due to high shear forces, EVs may aggregate or break, leading to reduced biological activity [106,107]. Ultracentrifugation is time-consuming and has low throughput, limiting its use to small-scale studies. Density gradient centrifugation (DGC) is more efficient for EV isolation which concentrates EVs in a discrete density band. However, the presence of residual gradient material may affect the purity of the finished product. Tangential flow filtration (TFF) is a gentle, size-based fractionation method for EV concentration. It is time-efficient and suitable for large-scale concentration. TFF provides higher EV yields and more effective removal of soluble proteins than enrichment by UC. EVs thus isolated are morphologically intact and display superior batch-to-batch consistency [72,108]. An MSC secretome concentrated by TFF was reported to yield higher protein, lipid, cytokine, and exosome contents than those concentrated by UC [109]. Size-exclusion chromatography (SEC) produces purer EVs. A previous study showed a 100-fold reduction in ferritin, a contaminant, in SEC-concentrated exosomes [110,111].

The enrichment of EVs by precipitation with polymers such as polyethylene glycol (PEG) inevitably leaves residual polymers in the preparation, interfering with downstream characterization and biological activities [111]. EVs prepared by the precipitation method have been shown to have lower biological activities than those enriched by DGC; this might be due to the masking of important EV surface molecules by residual precipitation reagents [111,112] and may render the EVs unusable for therapeutic applications. Moreover, this method is highly non-specific and can easily co-precipitate non-EV components. A combination of different enrichment methods may be used to enhance EV integrity and activities while reducing contamination. For example, SEC has been used to remove gradient material in DGC [113]. TFF combined with SEC has been reported to produce highly pure and functional EVs from large volumes of samples with minimal vesicle loss [114]. Details of the advantages and limitations of different enrichment methods can be found in a recent review [115].

The production and enrichment process, once standardized, should not be easily changed. Any modifications made to any step of the enrichment process need to be validated to confirm that the properties of the MSC-EVs remain unchanged.

### 4.2. MSC-EV Quality Control

Once the manufacturing procedures of MSC-EVs have been established, a list of QC standards to ensure the quality of the resulting EV preparations produced in different batches should be devised. While the characterization of MSCs and MSC-EVs should be as exhaustive as possible during the product development phase, the tests chosen for routine in-process control, recovery from storage and final product analysis should be technically relevant and feasible in terms of cost, labor and time. The quantity and identity, purity and impurities, sterility, potency, reproducibility, storage and formulation are the most commonly assessed parameters of MSC-EVs.

#### 4.2.1. Quality and Identity

##### Physicochemical Properties

The physical appearance of MSC-EVs, including the color, uniformity of mass and the presence of visible particles, needs to be noted and reported. General tests, such as pH and osmolality, of MSC-EV solutions are required. For freeze-dried MSC-EVs, the appearance of lyophilizate as well as the dissolution time, color, moisture content and clarity of the reconstituted solution should be recorded in the product development, in-process manufacturing and recovery phases, as well as in the final product.

##### Particle Size and Concentration

MSC-EV size and concentration, and optionally, zeta potential (ZP), should be documented in the product development, in-process manufacturing and recovery phases, as well as in the final product. The particle size of MSC-EVs can be determined by tunable resistive pulse sensing (TRPS), microfluidic resistive pulse sensing (MRPS), nanoparticle tracking analysis (NTA) or dynamic light scattering (DLS). Additionally, NTA, MRPS and TRPS make it possible to measure particle concentrations. Since these techniques cannot differentiate between EVs and non-EV particles and tend to overestimate EV concentration, unless antibodies staining is allowed, additional tests to validate the successful isolation of MSC-EV particles are needed. Many NTA devices can be used to analyze vesicles labelled with fluorescence dyes, which can increase the specificity of MSC-EV measurements. However, the sensitivity of such devices is limited for small particles. Moreover, only a subset of MSC-EVs can be measured, as there is no universal EV marker. Therefore, it is important to supplement this approach with examinations of MSC-EV morphologies by electron microscopy (EM) in the product development stage. The expression of MSC-EV quantities in cell or EV component (proteins, lipids, RNA) equivalents is helpful in dose-response studies and in gauging the yield of EVs in different batches of MSCs.

##### Zeta Potential

Zeta potential (ZP) describes the net electrical charge of molecules on the EV surface and is a measure of colloidal stability and aggregation of EVs. The particle environment (presence of charged or uncharged molecules that can adsorb on the particle surface, particle concentration and the pH and ionic strength of the solution) affects the ZP [116]. The ZP of MSC-EVs can be monitored using TRPS/MRPS/NTA/DLS and is expected to be slightly negative [117,118,119]. A stability study based on evaluating the colloidal behavior of MSC-EVs may be useful for identifying the optimal storage conditions.

##### Morphology

To differentiate EVs from non-EV particles, EM analysis is needed. Transmission electronic microscopy (TEM) shows the typical lipid bilayer of EV as a cup-shaped structure in the resulting image. It can show non-EV substances in the mixture, but it cannot quantify the amount of contaminating soluble factors. EM analysis may not be feasible as a routine test in the in-process manufacturing and recovery phases or in the final product but is recommended at least during the product development phase.

##### EV Phenotyping

EVs should be phenotyped by the expression of EV protein markers, as suggested by International Society of Extracellular Vesicles [86]. The minimal information for studies of EVs 2018 (MISEV2018) has suggested some criteria for defining EVs [86]. They recommended that at least one positive transmembrane/glycosylphosphatidylinositol (GPI)-anchored protein (e.g., tetraspanins (CD9, CD63, CD81)), one cytosolic protein recovered from EVs (e.g., ESCRT-I/II/III (TSG101) and the accessory protein ALIX, as well as heat shock proteins HSC70 and HSP84) and a negative non-EV protein marker in a specific system (e.g., apolipoprotein A1/2, apolipoprotein B, albumin for plasma) should be assessed. Quantitative analysis of potential contamination based on the presence of proteins in subcellular compartments other than the plasma membrane and endosomes, such as endoplasmic reticulum markers calnexin and GRP94/GP96, can also provide information about the purity of the exosomes. A list of potential contaminants for testing needs to be created in advance, based on the tissue source used for MSC-EV isolation.

Unlike the minimal criteria for defining multipotent MSCs proposed by the International Society for Cellular Therapy [93], the threshold percentage of particles expressing EV markers is not defined by MISEV [86]. Hence, Western blotting of EV markers is still commonly used to confirm EV identity. The result is semi-quantitative and requires large sample volumes, and, hence, is poorly adapted to high throughput analysis. Techniques such as enzyme-linked immunosorbent assay (ELISA), MACSPlex Exosome kit [120], small particle flow cytometry and nano-flow cytometry [121], which enable multiplexing and require only small amounts of EVs for analysis, are preferred for QC of MSC-EVs.

##### MSC-EV Content

As MSC-EVs carry inherently complex cell-specific cargos of proteins, lipids and genetic materials, standardized characterizations are challenging. In addition to the core signature of highly enriched vesicular proteins, other proteins reflecting the specific parent cell-origin and biogenesis pathway are present. From a therapeutic perspective, this complexity needs to be understood through comprehensive (multi)omic studies [122]. Mass spectrometry (MS)-based proteomics, metabolomics and lipidomics and next-generation sequencing transcriptomics (NGS/RNA-seq) allow high throughput and quantitative studies of EV-derived proteins, metabolites, lipids and RNA species, respectively [123]. These molecules can provide insights into unique biomarkers and the underlying biological mechanisms of MSC-EV-based therapeutics. This information is useful for the development of biomarkers for QC of MSC-EVs of different tissue origins and with different biogenesis pathways.

For routine QC in GMP facilities, it may not be suitable to use the non-targeted profiling approach described above for the measurement of proteins, metabolites, lipids and RNA species in MSC-EVs. As a proxy for the standardization of MSC-EV preparations, the use of simple, reliable and high throughput methods, such as bicinchoninic acid (BCA) assay for total proteins, sulfovanilin assay [124] or Nile red assay [109] for lipids and RiboGreen assay or UV-Vis spectrophotometry (e.g., NanoDrop) for RNA species, may be more appropriate, despite the fact that the sensitivities of these detection methods are limited. It is important to keep in mind that the total protein content is valid only if the MSC-EVs are isolated from cells cultured in a serum-free, chemically defined medium. Some RNA and DNA species may be presented in the MSC-EV preparation as un-encapsulated particles. We recommend measuring the RNA/DNA content in MSC-EV preparations with and without DNase/RNase treatment to differentiate RNA/DNA content in MSC-EVs from impurities in the development, manufacturing and recovery phases, as well as in the final product. For MSC-EV-based products with known RNA molecules, microarrays and real time quantitative reverse transcription polymerase chain reaction (qRT-PCR) can be used to test for the presence of specific RNA species in the development, manufacturing and recovery phase and in the final product.

#### 4.2.2. Purity and Impurities

##### Purity

Consensus on the definitions and quality metrics of EV purity remain to be established. The particle-to-protein ratio (P/µg protein) [125,126], protein-to-lipid ratio [124,127] and particle-to-RNA ratio [128] have been suggested as indicators of EV purity. Among them, the particle-to-protein ratio is commonly used, because impure preparations are expected to contain non-EV proteins, reducing the particle-to-protein ratio.

Based on the study of cell lines and biological fluids, Webber and Clayton [125] proposed the concentrations >3 × 10^10^ P/µg, 2 × 10^9^–2 × 10^10^ P/µg and <1.5 × 10^9^ P/µg to represent highly pure, less pure and impure EV preparations, respectively. However, it was demonstrated that it was difficult to reach the high purity ratio for biological fluids by simple ultracentrifugation and washing. The simple ultracentrifugation and washing of EVs derived from culture cells only reached the less pure ratio. More complex enrichment strategies or additional steps such as further concentration of EVs by DGC are required to achieve high purity; however, these methods may not be suitable for achieving the medium/high throughput required in clinical trials. Affinity capture-based approaches increase the purity of EVs, but the presence of antibodies may affect purity assessments. With the caveats that the particle-to-protein, protein-to-lipid and particle-to-RNA ratios are specific to a given biofluid or type of EV-producing cell and depend heavily on the respective enrichment method, they are useful in assessing batch-to-batch purity in a defined production system. These ratios should therefore be determined in-house for each specific MSC-EV product. Since the particle count obtained by nanoparticle tracking does not differentiate EV from non-EV particles, measuring the ratio of fluorescent EV particles expressing specific EV markers (CD9/CD63/CD81) to protein concentration may improve the specificity, although this method has not been validated.

It is understandable that a less pure MSC-EV preparation may raise concerns about efficacy and safety during clinical applications. While a certain degree of purity of MSC-EVs is definitely required for therapeutic applications, an extraordinarily pure MSC-EV preparation is not necessarily more effective than a less pure one. Additionally, highly pure MSC-EVs are not stable. Moreover, elevated concentration may damage the EVs and remove the loosely associated factors that act in conjunction with them, resulting in a loss of biological effects. Indeed, a recent study showed that the biological effects of MSC-EVs were actually attributable to the contaminating soluble factors present in the EV samples collected using a low-purity method [129]. Therefore, from the cost perspective of manufacturing MSC-EVs and where function is paramount, a less pure MSC-EV preparation may be more efficacious than a highly pure one. Some impurities, such as albumin and fibrinogen, are sometimes added to increase product stability. In cases where the impurities are not expected to have harmful effects on patients, batch-to-batch consistency is more important than reducing the level of impurities.

##### Impurities

Impurities can be process- or product-related. Process-related impurities include materials from the manufacturing steps, substrates or cell culture supplements. Product-related impurities include MSC-EVs degradation products which appear during manufacturing and storage. An upper limit of impurities should be defined for MSC-EV-based products. Measuring particle concentration and particle size, the concentration of biomarker-positive EVs and the total protein, total lipid and total RNA/DNA contents and their ratios will make it possible to indirectly monitor MSC-EVs and their degradation.

#### 4.2.3. Sterility

Standard microbiological tests determining endotoxin levels, microorganisms, mycoplasma and adventitious viruses, performed by certified laboratories, are mandatory for GMP-compliant MSC-EV manufacturing.

#### 4.2.4. Potency

In vitro biological assays that reflect the proposed/hypothesized mechanisms of action (MoA) of MSC-EVs in a quantitative manner should be developed for QC of the potency of different batches of MSC-EV preparations. Each therapeutic application of MSC-EVs should have its own in vitro potency assays to predict the intended therapeutic effects. Potency assays should be valid, reproducible, specific, sensitive, robust and cost-effective and have well-defined pass/fail criteria to meet the GMP-compliant release guidelines [130]. For instance, MSC-EVs have been shown to promote tendon and ligament healing by promoting tenogenesis, immunosuppression, proliferation and/or angiogenesis [131]. Hence, quantitative analysis of RNA or protein cargos that mediate these therapeutic effects may be one approach for quality control. Due to the complex MoA of MSC-EVs, the use of multiple assays to measure several MSC-EV attributes related to the intended use may be needed. Each individual potency assay should have its own reference standard, i.e., either a validated international or national one for established biochemical assays or one based results of an internally validated reference of MSC-EV preparations for cell-based assays.

#### 4.2.5. Reproducibility

Batch-to-batch reproducibility in the manufacturing process is a key parameter of every pharmaceutical drug. A previous study showed that the batch of secretome, among other factors, including lyophilization, the concentration of excipient and total amount of excipients, is the primary source of variability of the lipid and protein contents and the anti-elastase activity in the production of EVs derived from human adipose-derived stromal cells (ADSC-EVs) [132]. To reduce variability, MSC-EVs isolated from MSCs of different donors or different cell passages can be pooled to compensate for inter-donor differences. The quantity, identity, purity, impurities and potency of different batches of MSC-EVs can then be compared. The pool size is determined by the variability of samples from different donors or passages. Reductions in variability are theoretically the reciprocal of the square root of the number of samples pooled. This equation was confirmed in a previous study after pooling EVs harvested from peripheral blood mononuclear cells (PBMSCs) isolated from different donors [133]. The % deviation was calculated as the standard deviation (SD)/mean × 100%; this can give information about the variability of different batches. In a previous study, a % deviation of less than 10 was regarded as demonstrating low variation [133]. There is a potential safety concern regarding combining EVs from different donors. Strict donor selection criteria and in-process controls such as traceability of the donor should be implemented in the manufacturing process. To enhance traceability, the pooling of MSC-EVs derived from different cell stocks is preferred over the pooling of MSCs from different donors. As there are currently no standards for MSC-EV-based products, a batch developed in-house, shown to be stable and suitable for clinical trials, should be used as an internal reference by which to calibrate the results of different batches and analytical instruments. Potency relative to the in-house reference batch should be reported.

#### 4.2.6. Storage and Formulation

The storage and recovery conditions of isolated MSC-EVs can affect the EV characteristics, including stability, number of particles, aggregation and functions. The stability of biological products decreases over time during storage. The physiochemical and biological properties of MSC-EVs during storage for various times (e.g., 1, 3, 6 months) under accelerated and stress conditions (e.g., temperatures, relative humidity, pH) need to be determined to understand the degradation patterns and to define the limits of stability. EVs are sensitive to changes in pH, which can cause EV aggregation and loss of function. Isotonic buffers are recommended for storing EVs to prevent pH shifts during storage, freezing and thawing. Highly pure EVs may adhere to the surface of the storage container, making them impossible to recover. Low-protein binding synthetic materials should be used as storage containers. Common storage methods of MSC-EVs include freezing at −80 °C and lyophilization. The storage and retrieval conditions of MSC-EVs should be documented. If MSC-EV products are frozen, the number of repeated freeze–thaw cycles should be minimized to prevent EV aggregation, and QC measures should be built in after thawing in the manufacturing process. One or more cryoprotectants are usually added during cryopreservation [134]. Freeze-drying is expected to increase the ease of handling and shelf-life of MSC-EVs compared to suspensions. However, freeze-drying was reported to reduce the amount of proteins and lipids in ADSC-EV preparations [132]. The addition of lyoprotectants such as mannitol, trehalose or sucrose may improve the preservation of EVs under freeze-drying [134]. The choice of excipients can affect the protein and lipid contents, as well as the anti-elastase activity, of MSC-EV preparations [132,135,136] and should be tested. Since excipients affect QC procedures, they should be confirmed early in the manufacturing process.

The route of administration also influences the formulation, efficacy and safety of MSC-EVs. Biomaterials can be added to aid MSC-EV delivery. The kinetics of EV release from biomaterials should be determined. Local administration of MSC-EVs, ideally ultrasound-guided, is preferred, as tendon and ligament injuries are localized, allowing better control and fewer side effects of MSC-EV treatment. Surgeries such as ACLR and rotator cuff repair are assisted with arthroscopy to minimize damage to the surrounding healthy tissue and scarring, shorten recovery time and lower the risk of complications compared to open surgeries. Continuous flow of the irrigation buffer is required to visualize the operated site via an arthroscope. For arthroscopic-assisted tendon and ligament repair, MSC-EVs loaded in pre-formed biomaterial ex vivo are required. This avoids dispersion of MSC-EVs by the irrigation buffer and ensures their successful delivery to the operated site.

The QC panel for MSC-EV manufacturing is shown in Table 4.

### 4.3. Safety

Prior to application in clinical trials, the effect of dose and frequency of administration of MSC-EVs on toxicity and immunogenicity need to be evaluated in animals. Phase I clinical trials of pharmacodynamics, pharmacokinetics and potential off-target effects of MSC-EVs at the proposed route of administration are also needed. While MSC-EVs share the complexity of ATP products, they do not fulfill the definition of an ATP unless they contain the transgene-products from genetically modified cells. Despite this, the general scientific principles regarding quality (characterization, potency, reproducibility) and non-clinical and clinical requirements for pharmaceutical products are applicable to MSC-EVs [68]. Autologous MSC-EVs naturally occur in the body, and the release of MSC-EVs is a natural physiological process. Substances in MSC-EVs are also physiological body constituents. There is compelling evidence of a good safety profile of allogeneic MSCs in animal studies [140,141] and clinical trials [142,143]. Derived EVs are, therefore, expected to be safe. There is no evidence that allogeneic EVs cause any adverse events after transplantation in immunocompetent animals [144,145] (Table 2). EVs derived from ADSCs and BMSCs have been found to be non-immunogenic in an in vitro immunogenicity assay [109]. The transplantation of MSC-EVs, if derived from the same origin of target tissue, would further lower safety concerns. Recent clinical reports on the application of allogeneic MSC-EVs for the treatment of a human graft-versus-host disease patient [146], chronic kidney disease patients [147] and patients with COVID-19 [29] showed no adverse side effects, supporting their general safety.

### 4.4. Efficacy

The availability of animal data demonstrating the efficacy of MSC-EVs in tendon and ligament repair is crucial for subsequent clinical trials. MSC-EVs isolated from different tissues have been shown to promote healing of acute tendon injuries, acute ligament injuries, a failed healing model of tendinopathy and tendon-to-bone junction repair (Table 2). Further research is needed due to the different tissue origins, species, dose and route of administration of MSC-EVs, as well as the different animal models used in previous studies. Data on the dose-frequency response relationship of MSC-EVs are needed to determine the optimal dose for the treatment of specific tendon or ligament injuries. There is no consensus on the best normalization strategy for EV dose (e.g., number of cells or tissue mass), but a rationale should be provided, according to MISEV2018 [86]. As discussed, local injections of MSC-EVs appear to be more suitable for the treatment of local tendon and ligament injuries, and may reduce the dose and manufacturing cost of MSC-EVs in future clinical trials. One pre-clinical study reported systematic intravenous injection of 200 µg protein of BMSC-EVs weekly for the promotion of rotator cuff repair in a rat model [52]. The dose used was high and prohibited future clinical application.

Figure 1 outlines the manufacturing considerations and QC program for the development of GMP-grade MSC-EVs.

## 5. Conclusions

In recent years, there has been a boom in research on the effects of MSC-EVs for the treatment of various tendon and ligament injuries. However, none of this research has reached the clinical phase yet. Several key technical challenges need to be overcome for the clinical application of MSC-EVs. First, the biochemical composition of MSC-EVs remains unclear. The production or uptake mechanisms are poorly described, and GMP standards for clinical grade production, storage and recovery are lacking. Second, the drug loading efficiency of EVs is relatively lower than that of liposomes. Third, engineered MSC-EVs containing transgene products are categorized as ATPs, and hence, need to meet more stringent regulatory requirements. Fourth, as the half-life of MSC-EVs is short and they cannot replicate themselves like MSCs, large amounts of MSC-EVs are required for clinical application. The large scale and efficient production of MSC-EVs remains difficult at present. Finally, the physiochemical properties, particle size and concentration, as well as the MSC-EV content, have been the focus of numerous investigations. The MoA or potency of MSC-EVs remains relatively unexplored, resulting in a lack of appropriate functional assays. To overcome the bottleneck for MSC-EV-based therapies, clinical-grade MSCs meeting the requirements for therapeutic use and transplantation are needed for MSC-EV production. Conditions to ensure the sustainable release of MSC-EVs should be established. The specifications of MSC-EVs vary according to the enrichment method. Therefore, it is important to choose a cost-effective, scalable and automatable concentration method as early as possible to avoid changing the method and, hence, MSC-EV specification due to a subsequent scaling up of production. While there is some consensus regarding the transportation and storage of MSC-EVs at −80 °C, this poses challenges and is a cost-ineffective approach. Alternative methods, such as lyophilization, may improve MSC-EV stability during storage and transportation. The route of administration greatly influences the therapeutic efficacy and safety of MSC-EVs and should be determined prior to clinical application. Efforts to develop high-throughput and precise quantification methods for assessing MoA or potency will greatly facilitate the clinical translation of MSC-EV-based therapies. The product is the process. Any slight change in the MSC-EV production process, formulation or storage conditions can influence the composition and biological activity of the finished product. It is, hence, important to build in scalability, reproducibility and GMP concepts, along with QC measures and release criteria to MSC-EV-based products from the time of deciding to manufacture them into pharmaceutical products, or even earlier, at the pre-clinical stage. Researchers have to consider practical issues associated with manufacturing procedures and QC measures, as described in this review, in order to translate the pre-clinical results of MSC-EVs into clinical practice for the treatment of tendon and ligament injuries.

## Figures and Tables

**Figure 1 pharmaceutics-14-01684-f001:**
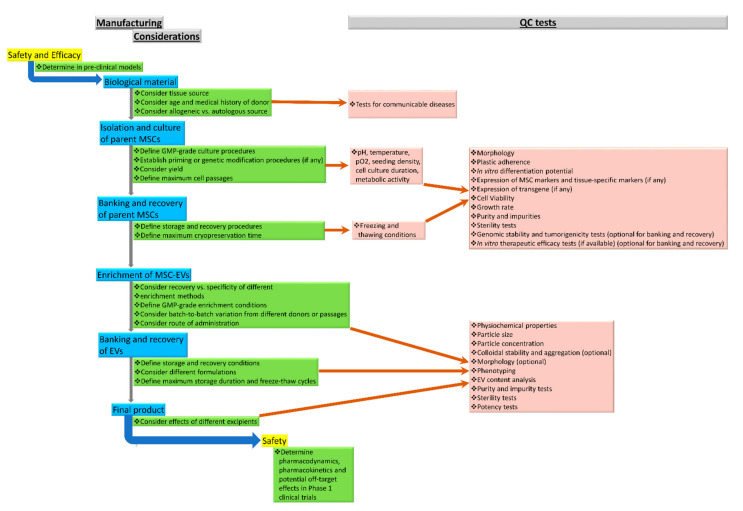
Manufacturing considerations and QC program for the development of GMP-grade MSC-EVs.

**Table 1 pharmaceutics-14-01684-t001:** List of clinical trials of MSC-EVs registered at www.clinicaltrials.gov (accessed on 30 June 2022).

Number	Study Title	Target Health Condition Estimated Enrollment of Patients Clinical Trial Phase	Type of EV Administrated Dosage Route of Administration	Status and Results (If Applicable)	Study Location	Reference/ Identifier
1	Bone Marrow Mesenchymal Stem Cell Derived Extracellular Vesicles Infusion Treatment for Acute Respiratory Distress Syndrome: A Phase I/II Clinical Trial	- ARDS - N = 81 - Phase I/II	- Allogeneic human BMSC-EVs (i.e., ExoFlo) - 10 mL or 15 mL of ExoFlo - IV	- Not yet recruiting - No result posted	Not specified	NCT05127122
2	Bone Marrow Mesenchymal Stem Cell Derived Extracellular Vesicles Infusion Treatment: A Global Expanded Access Protocol for Patients With COVID-19 Associated ARDS Who Do Not Qualify for Phase II Randomized Control Trial	- COVID-19 associated ARDS - Target number not provided - Expanded Access	- Allogeneic human BMSC-EVs (i.e., ExoFlo) - Dosage not specified - IV over 60 min	- Not specified - No result posted	Not specified	NCT04657458
3	Bone Marrow Mesenchymal Stem Cell Derived Extracellular Vesicles Infusion Treatment for COVID-19 Associated Acute Respiratory Distress Syndrome (ARDS): A Phase II Clinical Trial	- COVID-19 associated ARDS - N = 120 - Phase II	- Allogeneic human BMSC-EVs (i.e., ExoFlo) - 10 mL (8 × 10^11^ particles) or 15 mL (1.2 × 10^12^ particles) of ExoFlo in a total volume of 100 mL (mixed with 90 Ll or 85 mL of normal saline) - IV	- Completed - No result posted [29]	United States	NCT04493242
4	Bone Marrow Mesenchymal Stem Cell Derived Extracellular Vesicles as Early Goal Directed Therapy for COVID-19 Moderate-to-Severe Acute Respiratory Distress Syndrome (ARDS)	- COVID-19 associated ARDS - N = 400 - Phase III	- Allogeneic human BMSC-EVs (i.e., ExoFlo) - 15 mL of ExoFlo (1.2 × 10^12^) in 85 mL of saline on days 1 and 4 - IV over 60 min	- Not yet recruiting - No result posted	Not specified	NCT05354141
5	Bone Marrow Mesenchymal Stem Cell Derived Extracellular Vesicles Infusion Treatment for Mild-to-Moderate COVID-19: A Phase II Clinical Trial	- COVID-19 - N = 30 - Phase II	- Allogeneic human BMSC-EVs (i.e., ExoFlo) - 7 × 10^11^ to 10.5 × 10^11^ of particles once - IV	- Not yet recruiting - No result posted	Not specified	NCT05125562
6	ExoFlo™ Infusion for Post-Acute COVID-19 and Chronic Post-COVID-19 Syndrome	- Post-acute COVID-19 and chronic post-COVID-19 syndrome - N = 60 - Phase I/II	- Allogeneic human BMSC-EVs (i.e., ExoFlo) - 15 mL of ExoFlo, approximately equivalent to 10.5 × 10^8^ of particles (mixed with 85 mL of normal saline) - IV	- Not yet recruiting - No result posted	Not specified	NCT05116761
7	Intermediate Size Expanded Access for the Use of ExoFlo in the Treatment of Abdominal Solid Organ Transplant Patients Who Are at Risk of Worsening Allograft Function with Conventional Immunosuppressive Therapy Alone	- Organ transplant rejection - N = 20 - Expanded Access	- Allogeneic human BMSC-EVs (i.e., ExoFlo) - 15 mL of ExoFlo in a total volume of 100 mL (mixed with 85 mL of normal saline) up to nine times over 1 year - IV over 60 min	- Not yet recruiting - No result posted	Not specified	NCT05215288
8	A Phase I Study of ExoFlo, an ex Vivo Culture-expanded Adult Allogeneic Bone Marrow Mesenchymal Stem Cell Derived Extracellular Vesicle Isolate Product, for the Treatment of Medically Refractory Crohn’s Disease	- Refractory Crohn’s disease - N = 10 - Phase I	- Allogeneic human BMSC-EVs (i.e., ExoFlo) - 15 mL of ExoFlo on days 0, 2 and 4, weeks 2 and 6 and every 8 weeks thereafter to week 46 (n = 5) (total 10 doses) or 15 mL of ExoFlo on days 0, 2 and 4, weeks 2 and 6 and every 4 weeks thereafter to week 46 (n = 5) (total 15 doses) - IV	- Not yet recruiting - No result posted	Not specified	NCT05130983
9	A Phase I Study of ExoFlo, an ex Vivo Culture-expanded Adult Allogeneic Bone Marrow Mesenchymal Stem Cell Derived Extracellular Vesicle Isolate Product, for the Treatment of Medically Refractory Ulcerative Colitis	- Refractory ulcerative colitis - N = 10 - Phase I	- Allogeneic human BMSC-EVs (i.e., ExoFlo) - 15 mL of ExoFlo on days 0, 2 and 4, weeks 2 and 6 and every 8 weeks thereafter to week 46 (n = 5) (total 10 doses) or 15 mL of ExoFlo on days 0, 2 and 4, weeks 2 and 6 and every 4 weeks thereafter to week 46 (n = 5) (total 15 doses) - IV	- Not yet recruiting - No result posted	Not specified	NCT05176366
10	A Pilot Safety Study of the Administration of Mesenchymal Stem Cell Extracellular Vesicles in the Treatment of Burn Wounds	- 2nd degree burn wounds - N = 10 - Phase I	- Allogeneic human BMSC-EVs (i.e., AGLE-102) - EVs derived from approximately 1 × 10^4^ of BMSCs for each cm^2^2 treated area within 48 h of burn injury. Up to two additional administrations over a period of no more than 8 weeks. - Direct wound application	- Not yet recruiting - No result posted	Not specified	NCT05078385
11	A Safety Study of the Administration of Mesenchymal Stem Cell Extracellular Vesicles in the Treatment of Dystrophic Epidermolysis Bullosa Wounds	- Dystrophic epidermolysis bullosa wound - N = 10 - Phase I/IIA	- Allogeneic human BMSC-EVs (i.e., AGLE-102) - Dosage not specified; each administration will occur 14 days (±7 days) but no less than 7 days apart for a maximum of six administrations - Route of administration not specified	- Not yet recruiting - No result posted	Not specified	NCT04173650
12	A Safety Study of Intravenous Infusion of Bone Marrow Mesenchymal Stem Cell-derived Extracellular Vesicles (UNEX-42) in Preterm Neonates at High Risk for Bronchopulmonary Dysplasia	- Bronchopulmonary dysplasia - N = 3 (actually recruited) - Phase I	- Allogeneic human BMSC-EVs in PBS (i.e., UNEX-42) - 20 pmol, 60 pmol or 200 pmol phospholipid/kg body weight - IV	- Terminated due to business decision - No result posted	United States	NCT03857841
13	Effect of Adipose Derived Stem Cells Exosomes as an Adjunctive Therapy to Scaling and Root Planning in the Treatment of Periodontitis: A Human Clinical Trial	- Periodontitis - N = 10 - Early Phase I	- Autologous ADSC-Exos - Dosage not specified - Local injection	- Unknown status - No result posted	Egypt	NCT04270006
14	Study of Exosomes Derived from Mesenchymal Stem Cells on the Therapy for Children with Severe Infection	- Sepsis and critical illness - N = 200 - Phase not specified	- MSC-Exos - Dosage not specified - Route of administration not specified	- Not yet recruiting - No result posted	China	NCT04850469
15	A Pilot Clinical Study on Inhalation of Mesenchymal Stem Cells Exosomes Treating Severe Novel Coronavirus Pneumonia	- Novel coronavirus pneumonia (COVID-19) - N = 24 - Phase I	- Allogeneic human ADSC-Exos - 2.0 × 10^8^ of particles/3 mL on days 1, 2, 3, 4 and 5 - Aerosol inhalation	- Completed - Result published, not posted on ClinicalTrials.gov [28]	China	NCT04276987
16	Exosome of Mesenchymal Stem Cells for Multiple Organ Dysfunction Syndrome After Surgical Repair of Acute Type A Aortic Dissection: A Pilot Study	- Multiple organ failure - N = 60 - Phase not specified	- Allogeneic HUMSC-Exos - 150 mg of particles once a day for 14 days - IV	- Not yet recruiting - No result posted	Not specified	NCT04356300
17	Effect of Umbilical Mesenchymal Stem Cells Derived Exosomes on Dry Eye in Patients with Chronic Graft Versus Host Diseases	- Dry eye disease - N = 27 - Phase I/II	- Allogeneic HUMSC-Exos - 10 µg/drop four times a day for 14 days - Eye drop	- Recruiting - No result posted	China	NCT04213248
18	A Tolerance Clinical Study on Aerosol Inhalation of Mesenchymal Stem Cells Exosomes in Healthy Volunteers	- Healthy volunteers - N = 24 - Phase I	- Allogeneic human ADSC-Exos - 2.0 × 10^8^, 4.0 × 10^8^, 8.0 × 10^8^, 12.0 × 10^8^, or 16.0 × 10^8^ particles/3 mL once - Aerosol inhalation	- Completed, all volunteers tolerated the human ADSC-Exos nebulization well. No significant changes in vital signs (temperature, heart rate, respiratory rate and saturation oxygen) and laboratory parameters (alanine aminotransferase (ALT) level, creatinine level) were reported among volunteers in all groups during the nebulization or in the 7-day follow-up period [27]	China	NCT04313647
19	Mesenchymal Stem Cells Derived Exosomes Promote Healing of Large and Refractory Macular Holes	- Large and refractory macular holes - N = 44 - Early Phase 1	- Allogeneic HUMSC-Exos - 50 μg or 20 μg of particles in 10 μL of PBS - Intravitreal injection around MH	- Active, not recruiting - No result posted	China	NCT03437759
20	Phase 1 Study of The Effect of Cell-Free Cord Blood Derived Microvesicles On β-cell Mass in Type 1 Diabetes Mellitus (T1DM) Patients	- Diabetes Mellitus Type 1 - N = 20 - Phase I	- Allogeneic umbilical cord-blood-derived MSC exosomes and microvesicles - The first dose will be purified exosomes, ranging between 40–180 nm, in a dose of the supernatant produced from (1.22–1.51) × 10^6^/kg. The second dose, after 7 days, will be the microvesicles, ranging between 180–1000 nm, in a dose of the supernatant produced from (1.22–1.51) × 10^6^ /kg. - IV	- Status unknown - No result posted	Egypt	NCT02138331
21	Mesenchymal Stem Cell Exosomes for the Treatment of COVID-19 Positive Patients with Acute Respiratory Distress Syndrome and/or Novel Coronavirus Pneumonia	- COVID-19 with ARDS and/or novel coronavirus pneumonia - N = 55 - Phase I/II	- Allogeneic perinatal MSC-Exos - 2 × 10^9^, 4 × 10^9^ or 8 × 10^9^ particles per mL every other day for 5 days (three doses in total) for dose escalation study; Dose at 8 × 10^9^ particles per mL every other day for 5 days (three doses in total) for the double-blinded placebo-controlled randomized control trial - IV	- Not yet recruiting - No result posted	United States	NCT04798716
22	A Phase I Study Aiming to Assess Safety and Efficacy of a Single Intra-articular Injection of MSC-derived Exosomes (CelliStem^®^OA-sEV) in Patients with Moderate Knee Osteoarthritis (ExoOA-1)	- Knee osteoarthritis - N = 10 - Phase I	- Allogeneic MSC-Exos - 3–5 × 10^11^ particles/dose - Intra-articular knee injection	- Not yet recruiting - No result posted	Not specified	NCT05060107
23	A Phase II Trial to Investigate Clinical Efficacy of Autologous Synovial Fluid Mesenchymal Stem Cell-Derived Exosome Application in Patients with Degenerative Meniscal Injury	- Degenerative meniscal injury - N = 30 - Phase II	- Autologous synovial fluid-derived MSC-Exos - MSC-Exos derived from 1 × 10^6^ cells/kg - Intra-articular knee injection	- Recruiting - No result posted	Turkey	NCT05261360
24	The Protocol of Evaluation of Safety and Efficiency of Method of Exosome Inhalation in SARS-CoV-2 Associated Two-Sided Pneumonia	- COVID-19 pneumonia - N = 30 - Phase I/II	- MSC-Exos (i.e., EXO1 and EXO2) - EXO1: 0.5–2 × 10^10^ in 3 mL of solution at twice a day for 10 days EXO2: 0.5–2 × 10^10^ in 3 mL of solution at twice a day for 10 days - Aerosol inhalation	- Completed - No non-serious adverse events and serious adverse events at 30 days after clinic discharge. No adverse events using the inhalation procedures in 10 days, Preliminary results showed no difference in days of hospitalization, SpO_2_, serum C-reactive protein level and serum lactic acid dehydrogenase level of the exosome groups compared to the placebo groups.	Russia	NCT04491240
25	The Extended Protocol of Evaluation of Safety and Efficiency of Method of Exosome Inhalation in COVID-19 Associated Two-Sided Pneumonia	- COVID-19 pneumonia - N = 90 - Phase II	- MSC-Exos (i.e., EXO1 and EXO2) - EXO1: 0.5–2 × 10^10^ particles in 3 mL of solution at twice a day for 10 days; EXO2: 0.5–2 × 10^10^ particles in 3 mL of solution at twice a day for 10 days - Aerosol inhalation	- Enrollment by invitation - No result posted	Russia	NCT04602442
26	A Multiple, Randomized, Double-blinded, Controlled Clinical Study of Allogeneic Human Mesenchymal Stem Cell Exosomes (hMSC-Exos) Nebulized Inhalation in the Treatment of Acute Respiratory Distress Syndrome	- ARDS - N = 169 - Phase I/II	- Allogeneic human MSC-Exos - 2.0 × 10^8^, 8.0 × 10^8^ or 16.0 × 10^8^ particles on days 1, 2, 3, 4, 5, 6 and 7 - Aerosol inhalation	- Recruiting - No result posted	China	NCT04602104
27	A Clinical Study of Allogeneic Human Adipose-derived Mesenchymal Progenitor Cell Exosomes (haMPC-Exos) Nebulizer for the Treatment of Carbapenem-resistant Gram-negative Bacilli-induced Pulmonary Infection	- Pulmonary infection caused by drug-resistant bacteria - N = 60 - Phase I/II	- Mesenchymal progenitor cells (MPCs)-derived Exos - 8.0 × 10^8^ or 16.0 × 10^8^ particles on days 1, 2, 3, 4, 5, 6 and 7 - Aerosol inhalation	- Recruiting - No result posted	China	NCT04544215
28	Open-Label, Single-Center, Phase I/II Clinical Trial to Evaluate the Safety and the Efficacy of Exosomes Derived from Allogeneic Adipose Mesenchymal Stem Cells in Patients with Mild to Moderate Dementia Due to Alzheimer’s Disease	- Alzheimer’s disease - N = 9 - Phase I/II	- Allogeneic ADSC-Exos - 5 μg or 10 μg or 20 μg of particles in 1 mL, twice per week for 12 weeks - Nasal drip	- Recruiting - No result posted	China	NCT04388982
29	The Use of Exosomes in Craniofacial Neuralgia	- Craniofacial neuralgia - N = 100 - Phase not specified	- Neonatal stem cell-derived Exos - 3 mL of particles (15 mg) delivered by ultrasound-guided regional epineural injection or 3 mL of particles (45 mg) delivered IV - IV without or without focused ultrasound-enhanced delivery	- Suspended - No result posted	United States	NCT04202783
30	Phase I Study of Mesenchymal Stromal Cells-Derived Exosomes with KrasG12D siRNA for Metastatic Pancreas Cancer Patients Harboring KrasG12D Mutation	- Pancreatic cancer with KrasG12D mutation - N = 28 - Phase I/II	- MSC-Exos loaded with Kras G12D siRNA - Infusion for 15–20 min on days 1, 4, and 10; Treatment repeats every 14 days for up to three courses, responders will be treated with three additional courses - IV	- Recruiting - No result posted	United States	NCT03608631
31	Safety and Efficacy of Allogeneic Mesenchymal Stem Cells Derived Exosome on Disability of Patients with Acute Ischemic Stroke: A Randomized, Single-blind, Placebo-controlled, Phase 1, 2 Trial	- Acute ischemic stroke - N = 5 - Phase I/II	- Allogeneic MSC-Exos enriched with miR-124 - Dosage not specified - Stereotaxis/Intraparenchymal	- Recruiting - No result posted	Iran	NCT03384433
32	Safety and Efficacy of Injection of Human Placenta Mesenchymal Stem Cells Derived Exosomes for Treatment of Complex Anal Fistula	- Refractory anal fistula - N = 80 - Phase I/II	- Allogeneic placental MSC-Exos - Dosage not specified, weekly for 3 weeks - Injection in fistula tract	- Recruiting - No result posted	Iran	NCT05402748
33	Efficacy and Safety of EXOSOME-MSC (Mesenchymal Stem Cell-Derived Exosomes) Therapy to Reduce Hyper-inflammation In Moderate COVID-19 (2019- New Corona Virus Disease) Patients	- COVID-19 - N = 60 - Phase I/II	- MSC-Exos - Dosage not specified, on days 1 and 7 - IV	- Recruiting - No result posted	Indonesia	NCT05216562
34	Exosome-based Nanoplatform for Ldlr mRNA Delivery in Familial Hypercholesterolemia	- Homozygous familial hypercholesterolemia - N = 30 - Phase I/II	- BMSC-Exos loaded with low-density lipoprotein (LDL) receptor (Ldlr) mRNA - Dose escalation phase: single dose of 0.044 mg/kg, 0.088 mg/kg, 0.145 mg/kg, 0.220 mg/kg, 0.295 mg/kg and 0.394 mg/kg extension phase: three intravenous/peritoneal infusion treatment once a week for three weeks - IV or IP	- Not yet recruiting - No result posted	China	NCT05043181
35	The Effect of Wharton Jelly-derived Mesenchymal Stem Cells and Stem Cell Exosomes on Visual Functions in Patients with Retinitis Pigmentosa	- Retinitis pigmentosa - N = 135 - Phase II/III	- Wharton jelly-derived MSC and their Exos - Dosage not specified - Subtenon injection	- Not yet recruiting - No result posted	Not specified	NCT05413148

No clinical studies on MSC-EV were found in clinicaltrialsregister.eu (accessed on 30 June 2022), with the keyword of “Extracellular vesicle”, “exosome”, “EV”, “micro-vesicle” and “nano-vesicle”. Abbreviations: bone marrow-derived stromal cells (BMSCs), adipose-derived stromal cells (ADSCs), mesenchymal stromal cells (MSCs), human umbilical cord mesenchymal stromal cells (HUMSCs), extracellular vesicles (EVs), exosomes (Exos), phosphate buffered saline (PBS), intravenous (IV), intraperitoneal infusion (IP); Acute Respiratory Distress Syndrome (ARDS).

**Table 2 pharmaceutics-14-01684-t002:** Summary of pre-clinical studies investigating the therapeutic effects of MSC-EVs on tendon and ligament repair.

Animal Model	EV Source Dosage Frequency Route of Administration	Follow Up Time Point	Results	References
Achilles tendon transection and repair in nude mouse	- BMSC-EVs educated macrophages (EEM) (3 × 10^9^ human BMSC-derived EVs were used to educate 75 cm^2^ flask of human macrophages to M2 phenotype for 3 days) - 1 × 10^6^ of EEM in 20 µL of saline; same amount of BMSC-EVs used to educate macrophage was also tested but exact dosage not reported - Immediately after repair, once - Injection in the surgical pouch around the injured tendon	7 and 14 days post-injury	EEM treatment substantially improved the biomechanical properties of the healing tendon but showed no improvement in collagen fiber organization. The EV or BMSC treatment showed biological responses but no effects on the biomechanical properties of tendon or collagen fiber organization. Both EEM and EV treatment reduced the MI/M2 ratio. EV, but not EEM, further increased the number of endothelial cells compared to injury only and EEM treatments. Although both EEM and EV treatments reduced the protein expression of collagen type I, no treatment effects were observed with the expression of type III collagen, type I/type III collagen ratio or collagen organization.	[38]
Mouse Achilles tendon two-third partial transection and repair	- Mouse naïve and IFNγ-primed ADSC-EVs laden on collagen sheet - 5–6 × 10^9^ of particles derived from 5 × 105 of ADSCs - Once - EV-laden collagen sheet was wrapped around injury site	1, 3 and 7 days post-injury	Compared with the untreated control group, primed ADSC-EVs, but not their unprimed counterparts, further reduced the rate of post-repair tendon gap formation and rupture and promoted collagen formation at the injury site. Primed ADSC-EVs, but not unprimed EVs, attenuated the early tendon inflammatory response after injury via modulation of the macrophage inflammatory response.	[39]
Rat mid-Achilles tendon transection and suture repair	- HUMSC-Exos - 200 μg of particles in 50 μL of PBS - Once - Subcutaneous injection	3 weeks post injury	HUMSC-Exos relieved tendon adhesion in rats when compared to PBS. The degree of inflammatory infiltration was lower in the HUMSC-Exos group compared to the PBS and sham groups. HUMSC-Exos significantly decreased COL III, α-SMA, p-p65 and COX2 expression.	[40]
Rat Achilles tendon transection and repair	- Rat hydroxycamptothecin-primed umbilical cord stem cells-derived EVs (HCPT-EVs) - 200 μg of particles in 50 μL of PBS - Once - Injection at the injury site after wound closure	3 weeks post-injury	Both HCPT-EVs and unprimed EVs reduced tendon adhesion. However, only HCPT-EVs significantly improved the histological healing score. There was no significant improvement in the maximal tensile strengths of the healing tendon after treatment with HCPT-EVs or unprimed EVs. HCPT-EVs contained more endoplasmic reticulum stress (ERS)-associated protein compared to unprimed EVs and activated the ERS pathway in fibroblast to counteract myofibroblast differentiation.	[41]
Rat mid-Achilles tendon transection and suture repair	- HUMSC-Exos - 100 μg of particles dissolved in 50 μL of PBS - Once - Subcutaneous injection	7 days post-injury	HUMSC-Exos promoted tendon repair via exosomal miR-27b-3p, which increased cell proliferation, invasion and RhoA activity of primary injured tenocytes.	[42]
Rat Achilles tendon window injury	- Rat BMSC-EVs - 2.8 × 10^12^ or 8.4 × 10^12^ of particles in 50 µL of PBS - Once - Injection at the injury site after wound closure	30 days post-injury	BMSC-EVs accelerated tendon repair in a dose-dependent manner. Higher doses of BMSC-EVs resulted in better restoration of tendon architecture, improved tendon-fiber alignment and lower vascularity. Higher concentrations of EVs induced higher expression of collagen type I and lower expression of collagen type III compared to the PBS control group and BMSC group.	[43]
Rat Achilles tendon central one-third window injury	- Rat Achilles tendon TDSC-Exos - 200 μg of particles in 30 μL of GelMA - Once - Local application	1, 2 and 8 weeks post-injury	TDSC-Exos promoted tendon repair by improving collagen fiber alignment and diameter, as well as inhibiting inflammation, accumulation of apoptotic cells and scar formation.	[44]
Rat Achilles rectangular full-thickness defect	- HUMSC-Exos - 100 μg of particles in 50 μL of fibrin glue - Once - Local application	2 and 4 weeks post-injury	HUMSC-Exos accelerated tendon healing via exosomal miR-29a-3p-mediated activation of PTEN/mTOR/TGF-β1 signaling pathway.	[45]
Rat patellar tendon window injury	- Rat BMSC-EVs - 25 µg of particles in 10 µL of fibrin glue - Once - Local application	2 and 4 weeks post-injury	BMSC-EVs promoted tendon healing with improvement in collagen fiber alignment, expression of tendon matrix genes and tenogenic differentiation markers compared to the fibrin glue-only group and untreated group. Inflammation and accumulation of apoptotic cells were suppressed, while the numbers of tendon progenitor cells increased at the healing site.	[46]
Rat patellar tendon window injury	- Rat BMSC-Exos in fibrin glue - 20 µg of particles in 10 µL of fibrin glue - Once - Local application	3 days and 1, 2, 4 weeks post-injury	BMSC-Exos improved the histological scores, promoted the proliferation of resident tendon stem cells and enhanced the expression of tendomodulin and type I collagen, as well as the biomechanical properties of neotendon.	[47]
Rat patellar tendon central one-third window injury	- Rat ADSC-Exos - 200 μg of particles in 30 μL of GelMA - Once - Local application	7, 14 and 28 days post-injury	Rat ADSC-Exos promoted tendon repair by improving the alignment of collagen fibers. The gene expression of TNMD, collagen I and SCXA, as well as the CD146^+^ TSCs at the injury site, increased significantly in the ADSC-derived exosome group.	[48]
Rat patellar tendon central one-third window injury	- Rat patella TDSC-Exos - 100 μg/mL of particles in 50 mg/mL of photopolymerizable hyaluronic acid (p-HA) - Once - Local application	2, 4 and 8 weeks post injury	pHA-TDSC-Exos promoted tendon healing with improvement in histology and biomechanical properties compared to the control group. TDSC-Exos enhanced tendon repair through miR-144-3p-regulated tenocyte proliferation and migration.	[49]
Rabbit chronic rotator cuff transection and tendon–bone repair	- ADSC-Exos - 10^11^ of particles in 20 µL of saline - Once - Local injection	18 weeks post-injury	Exos reduced fatty infiltration, increased the histological score with more fibrocartilage and improved biomechanical properties of the tendon–bone junction compared to the saline group.	[50]
Rabbit supraspinatus tendon injury	- Rabbit BMSC-Exos wrapped with polyaspartic acid-polylactic acid (PASP-PLA) microcapsules supplemented with BMP-2 - specified dose - Once - Local application	6-, 12-, and 18-weeks post-injury	Exos wrapped with “BMP-2 supplemented PASP-PLA microcapsules” promoted tendon and bone interface healing after rotator cuff injury via the Smad/RUNX2 signaling pathway. The expressions of tendon regeneration- and cartilage differentiation-related proteins were significantly upregulated.	[51]
Rat rotator cuff repair	- Rat BMSC-Exos - 200 μg of particles in 200 μL of PBS - Once - Systematic tail vein injection	4- and 8-weeks post-injury	BMSC-Exos promoted tendon–bone healing. It reduced the serum level of pro-inflammatory cytokines and inhibited the expression and distribution of M1 macrophages. The biomechanical properties and histology of the tendon–bone interface were also improved.	[52]
Rat supraspinatus tendon injury and repair	- Human ADSC-Exos - 30 μg of particles in 100 μL of hydrogel - Once - Local application	4- and 8-weeks post-injury	The ADSC-Exos-hydrogel group promoted rotator cuff repair compared to the control group, with improved histology and biomechanical properties.	[53]
Rat supraspinatus tendon transection and repair	- Blood derived purified exosome product (i.e., PEP) - 3.8 × 10^10^ of particles per mL in 3 × 3 × 3 mm of TISSEEL - Once - Local application	6 weeks post-injury	PEP in TISSEEL increased the mRNA expression of Col1, Col3, Scx, Tnmd, Tnc, Dcn and IGF compared to the control group. It also promoted remodeling of collagen fibers and new cartilage-like tissue formation at the tendon–bone interface after 6 weeks.	[54]
Mouse Achilles tendon–bone reconstruction model	- Mouse BMSC-Exos in hydrogel - dosage not reported - Once - Local application at the bone tunnel	7 and 14 days and 1 month post-injury	Mouse BMSC-Exos enhanced cell proliferation and reduced apoptosis at the injury site. It also increased the formation of fibrocartilage and improved M2 macrophage polarization and the biomechanical properties of the tendon–bone interface.	[55]
Mouse extra-articular Achilles tendon–bone tunnel model	- Exosomes derived from mouse Scx overexpressing PDGFRα(+) BMSCs - 1 × 10^10^ of particles (direct injection to the bone tunnel with tendon graft) - Once - Local injection into bone tunnel and incubated for 3–5 min for absorption before graft insertion	1, 2 and 3 weeks post-injury	Exosomes derived from Scx overexpressing PDGFRα(+) BMSCs reduced osteoclastogenesis and improved tendon–bone healing strength via exosomal miR-6924-5p.	[56]
Rat collagenase induced Achilles tendinopathy	- Rat Achilles tendon TDSC-Exos - Dosage not reported - Twice a week for 4 weeks starting one-week post-injury - Local injection	5 weeks post-injury	TDSC-Exos promoted tendon repair both histologically and biomechanically compared to the injury group; the effects were comparable to TDSC treatment.	[57]
Rat collagenase-induced Achilles tendinopathy	- Rat TDSC-Exos - No specified dose - Once - Exosomes wrapped in nitric oxide nanomotor and delivered via microneedle array	14 days post-injury	TDSC-Exos promoted healing of collagenase-induced Achilles tendinopathy via enhancing tendon cell proliferation, increasing the expression of Col1a, suppressing inflammation and preventing extracellular matrix degradation.	[58]
Rat carrageenan-induced quadriceps tendon tendinopathy	- Small EVs releasecd from human induced pluripotent stem cell-derived mesenchymal stem cells (i.e., iMSC-sEVs) - 1 × 10^10^ particles of iMSC-sEVs in 100 μL of PBS - Starting one week after carrageenan-induced injury, once a week for 4 weeks - Local injection	4 weeks after model establishment	iMSC-EVs alleviated pain and histologically improved tendinopathy characteristics by increasing cell proliferation and downregulating genes involved in inflammation and collagen degeneration.	[59]
Rat medial collateral ligament transection and repair	- Human BMSC-EVs and BMSC-EVs educated macrophages (i.e., EEMs) (3 × 10^9^ of human BMSC-derived EVs were used to educate 75 cm^2^2 flask of human macrophages to M2 phenotype for 3 days) - 1 × 10^9^ of particles in PBS or 20 μL of 1 × 10^6^ of EEMs - Once - Local application	14 days post-injury	Both BMSC-EVs and EEMs. BMSC-EVs promoted ligament repair with improvement in collagen (type I and III) production and collagen organization, as well as reduction of scar formation. EMMs improved the mechanical properties of healing ligament and reduced the M1/M2 macrophage ratio.	[60]
A case study of a horse suffering from suspensory ligament injury	- Allogeneic microvesicles (MVs) derived from 5-azacytidine (AZA) and resveratrol (RES)-treated ADSCs isolated from horse with metabolic syndrome - Dosage not reported - Twice (7 days after injury and 9 months after first injection) - Ultrasound-guided injection into injury site	10 months and 12 months after first injection	MVs improved the lesion filling, angiogenesis and elasticity of injured tissue.	[61]

Abbreviation: extracellular vesicle (EVs), exosomes (Exos), bone marrow-derived stromal cells (BMSCs), adipose-derived stromal cells (ADSCs), human umbilical cord mesenchymal stem cells (HUMSCs), phosphate buffered saline (PBS), tendon-derived stem cells (TDSCs).

**Table 3 pharmaceutics-14-01684-t003:** Suggested QC panel of parent MSC bank for EV production [69,91,92,93,101,102,103,104,105].

				Phase (1)		
				Developmental Stage	Clinical Batch Production	
Assessment Category	Assessment Items	Method	Release Criteria		Cell Production (2)	Thawing and Recovery after Cryopreservation
Cell culture conditions	pH, temperature, pO_2_, seeding density, cell culture duration, metabolic activity	Recording from cell incubator or bioreactor	Defined in-house	Mandatory	Mandatory	Not applicable
	Maximum cell passages	Not applicable	Defined in-house	Mandatory	Mandatory	Not applicable
Cryopreservation and thawing procedures	Cryopreservation and thawing methods, maximum cryopreservation time	Developed in-house	Defined in-house	Mandatory	Not applicable	Mandatory
Identity	Morphology	Microscopic observation	Defined in-house	Mandatory	Mandatory	Mandatory
	Plastic adherence	Microscopic observation	Adherent to plastic in standard culture conditions [93]	Mandatory	Mandatory	Mandatory
	In vitro differentiation potential	Tri-lineage differentiation kit	Differentiate into osteoblasts, adipocytes and chondrocytes as shown by staining of in vitro cell culture [93]	Mandatory	Mandatory	Mandatory
	Expression of MSC markers	Flow cytometry	According to ISCT criteria, ≥95% of cells positive for CD90, CD105, CD73 ≤2% of stained cells positive for CD45, CD34, CD14 or CD11b, CD79a or CD19, HLA-DR [93] Vary with MSC type	Mandatory	Mandatory	Mandatory
	Expression of tissue-specific markers (if any)	Defined in-house	Defined in-house	Recommended	Recommended	Recommended
	Expression of transgene (if any)	Defined in-house	Defined in-house	Mandatory	Mandatory	Mandatory
Viability and cell proliferation	Live/dead cell population	Cell counting/trypan blue dye exclusion	≥80% cell viability for routine culture, ≥70% cell viability after cell recovery [91,92]	Mandatory	Mandatory	Mandatory
	Growth rate	Calculation of population of doubling time and population doubling level (3)	Defined in-house	Mandatory	Mandatory	Mandatory
Purity and impurities (4)	Residual chemicals and biologicals (e.g., cell priming molecules, vector of transgene, FBS, serum proteins such as albumin, fibrinogen, cryoprotectants)	Specific for a given biofluid or tissue source for MSC isolation, MSC isolation method or preservation method	Defined in-house	Mandatory	Mandatory	Mandatory
Sterility	Endotoxin	Limulus amebocyte lysate (LAL) test	<0.5 EU/mL	Mandatory	Mandatory	Mandatory
	Bacteria	Direct inoculation	No growth of microorganisms	Mandatory	Mandatory	Mandatory
	Fungi	Direct inoculation	No growth of microorganisms	Mandatory	Mandatory	Mandatory
	Mycoplasma	PCR	Negative	Mandatory	Mandatory	Mandatory
	Adventitious virus	In vitro adventitious viral agent test	Negative	Mandatory	Mandatory	Mandatory
Stability and safety	Genomic stability	Giemsa-banded karyotyping Comparative genomic hybridization Fluorescence in situ hybridization Genome sequencing	Absence of chromosomal and genomic abnormalities	Mandatory	Mandatory	Optional
	Tumorigenicity	In vitro soft agar colony formation assay	Absence of cell colonies	Mandatory	Mandatory	Optional
Potency (5)	In vitro test of therapeutic efficacy of MSCs (if available)	Developed in-house	Defined in-house	Optional	Optional	Optional

(1) Categorized as Mandatory, Recommended, Optional, Not applicable. (2) Cell production can be managed as different tiers: master cell bank (MCB), working cell banks (WCB) and post-production cell bank (PPCB). The tests required vary with the tier of the cell banking system; (3) Population doubling time (PDT) = t × log (2)/log (number of cells harvested/number of cells plated), where t is the time in hours between passage 1 and cell harvest. Population doubling level (PDL) = 3.322 (log Y − log I), where Y = number of cells harvested and I = number of cells plated at P1. (4) Purity of MSC culture is also indicated by the expression of positive and negative MSC markers. (5) As the MSCs are used for EV production, the potency test can be done at the QC of EV production. Abbreviation: fetal bovine serum (FBS), polymerase chain reaction (PCR).

**Table 4 pharmaceutics-14-01684-t004:** Suggested QC panel of MSC-EVs at different stages of product development [68,69,86,87,101,109,115,124,125,126,127,128,137,138,139].

				Phase (1)			
				Development Stage	**Clinical Batch Production**		
Assessment Category	Assessment Items	Method	Release Criteria		In-Process Manufacturing Stage	Recovery Stage	Final Product
**Quantity and identity**							
Physiochemical properties	For EV in solution						
	pH	pH meter	Defined in-house	Mandatory	Mandatory	Mandatory	Mandatory
	Osmolality	Osmometer	Defined in-house	Mandatory	Mandatory	Mandatory	Mandatory
	Color	Physical appearance examination	Defined in-house	Mandatory	Mandatory	Mandatory	Mandatory
	Mass uniformity	Balance	Defined in-house	Not applicable	Not applicable	Not applicable	Mandatory
	Presence of visible particles	Physical appearance examination	Defined in-house	Mandatory	Mandatory	Mandatory	Mandatory
	For freeze dried EV						
	Appearance of lypoilisate	Physical appearance examination	Defined in-house	Mandatory	Mandatory	Mandatory	Mandatory
	Solubility	Dissolution time	Defined in-house	Mandatory	Mandatory	Mandatory	Mandatory
	Color	Physical appearance examination	Defined in-house	Mandatory	Mandatory	Mandatory	Mandatory
	Moisture content	Weight difference after drying	Defined in-house	Mandatory	Mandatory	Mandatory	Mandatory
	Clarity of reconstituted solution	Physical appearance examination	Defined in-house	Mandatory	Mandatory	Mandatory	Mandatory
Particle size and concentration	Particle size range	TRPS/MRPS/NTA/DLS	Defined in-house	Mandatory	Mandatory	Mandatory	Mandatory
	Particle concentration	TRPS/MRPS/NTA	Defined in-house	Mandatory	Mandatory	Mandatory	Mandatory
Colloidal stability and aggregation	Zeta potential (surface charge)	TRPS/MRPS/NTA/DLS	Defined in-house	Optional	Optional	Optional	Optional
Morphology	Structure	Electron Microscopy	Cup-shaped structure	Mandatory	Recommended	Optional	Optional
Phenotyping	Positive EV marker	WB, ELISA, MASCPlex Exosome Kit, nano/small particle flow cytometry	Defined in-house; at least one positive GPI-anchored protein (e.g., CD9, CD63, CD80) and one cytosolic protein (e.g., TSG101, ALIX, HSC70, HSP84) according to MISEV2018 [86]	Mandatory	Mandatory	Mandatory	Mandatory
	Negative non-EV marker	WB, ELISA, MACSPlex Exosome Kit, nano/small particle flow cytometry	Defined in-house; at least one negative marker depending on tissue source according to MISEV 2018. Examples are apolipoprotein A1/2, apolipoprotein B, albumin for plasma, endoplasmic reticulum markers, Golgi markers [86]	Mandatory	Mandatory	Mandatory	Mandatory
EV content	Multi-omics study	Proteomics, metabolomics, lipidomics, transcriptomics	For exploratory purpose, no release criteria	Recommended	Not applicable	Not applicable	Not applicable
	Protein concentration (2)	BCA protein assay	Defined in-house	Mandatory	Mandatory	Mandatory	Mandatory
	Lipid concentration (2)	Sulfovanilin assay/ Nile red assay	Defined in-house	Mandatory	Mandatory	Mandatory	Mandatory
	DNA/RNA concentration (2)	UV-Vis spectrophotometry (with or without Rnase/Dnase treatment)	Defined in-house	Mandatory	Mandatory	Mandatory	Mandatory
**Purity and impurities** (3)							
	Particle-to-protein ratio [125,126] (4)	Refer to “Particle concentration” and “Protein concentration”	Defined in-house; Highly pure: >3 × 10^10^ P/μg, Less pure: 2 × 10^9^–2 × 10^10^ P/μg, Impure: <1.5 × 10^9^ P/μg [125]	Mandatory	Mandatory	Mandatory	Mandatory
	Protein-to-lipid ratio [124,127] (4)	Refer to “Protein concentration” and “Lipid concentration”	Defined in-house	Mandatory	Mandatory	Mandatory	Mandatory
	Particle-to-RNA ratio [128] (4)	Refer to “Particle concentration” and “DNA/RNA concentration”	Defined in-house	Mandatory	Mandatory	Mandatory	Mandatory
	Residual chemicals and biologicals (e.g., cryoprotectants, lysoprotectants, residual priming molecules, serum proteins if EV collection in complete culture medium, residual chemicals of EV enrichment)	Specific for a given biofluid or type of EV-producing cell, EV enrichment method or preservation method	Defined in-house	Mandatory	Mandatory	Mandatory	Mandatory
	Excipients	Specific to a given storage method or route of administration	Defined in-house	Mandatory	Mandatory	Mandatory	Mandatory
**Sterility**							
	Endotoxin	Limulus amebocyte lysate (LAL) test	<0.5 EU/mL	Mandatory	Mandatory	Mandatory	Mandatory
	Bacteria	Direct inoculation	No growth of microorganisms	Mandatory	Mandatory	Mandatory	Mandatory
	Fungi	Direct inoculation	No growth of microorganisms	Mandatory	Mandatory	Mandatory	Mandatory
	Mycoplasma	PCR	Negative	Mandatory	Mandatory	Mandatory	Mandatory
	Adventitious virus	In vitro adventitious viral agent test	Negative	Mandatory	Mandatory	Mandatory	Mandatory
**Potency**							
	Presence of specific RNA, proteins or lipids or in vitro activity assay(s) important for therapeutic functions (if available)	Developed in-house	Defined in-house	Mandatory	Mandatory	Mandatory	Mandatory

(1) Categorized as Mandatory, Recommended, Optional, Not applicable. (2) At least one of these indicators. (3) Purity of Evs is also indicated by the expression of negative expression of non-EV markers. (4) At least one of these indicators. Abbreviation: tunable resistive pulse sensing (TRPS), microfluidic resistive pulse sensing (MRPS), nanoparticle tracking analysis (NTA), dynamic light scattering (DLS), transmembrane/glycosylphosphatidylinositol (GPI)-anchored protein, polymerase chain reaction (PCR), Western blotting (WB), enzyme-linked immunosorbent assay (ELISA).

## Data Availability

Not applicable.

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
