# Peer review of "Practical Considerations for Translating Mesenchymal Stromal Cell-Derived Extracellular Vesicles from Bench to Bed"

_pharmaceutics, 2022, doi:10.3390/pharmaceutics14081684_

Round 1

Reviewer 1 Report

In this manuscript entitled Practical Considerations for Translating Mesenchymal Stromal 2 Cell-derived Extracellular Vesicles from Bench to Bed, the authors summarized the Extracellular vesicles (EVs) derived from mesenchymal stromal cells (MSCs) have shown potential for the treatment of tendon and ligament injuries. I also think that the topic may be of interesting to the field. However, the inaccuracies in this paper are so numerous that I cannot list all of them within the usual review period. The following points should be addressed for paper re-submission. I haven't started the review yet.

Comments:

1.     There are many errors or insufficient explanations in the manuscript. Below are the examples: “discuss about”, “fulfil” The authors should carefully check the text. Correction of the paper by a native English speaker is recommended.

2.     The papers cited in the Table are not listed in References.

3.     1.1: Why did you only present the examples of United States and New Zeeland? Please correct the sentence as it is difficult for the reader to read.

4.     1.1: This is not an introduction that describes Stem-cell based therapy.

It is difficult to continue peer review due to many careless mistakes and fatal errors in logical composition.

Author Response

  1. There are many errors or insufficient explanations in the manuscript. Below are the examples: “discuss about”, “fulfil” The authors should carefully check the text. Correction of the paper by a native English speaker is recommended.

Reply: We are sorry for the typos and grammatical mistakes. We have carefully checked and revised the manuscript.

  1. The papers cited in the Table are not listed in References.

Reply: The papers cited in the Table are included in the references in the revised manuscript.

  1. 1.1: Why did you only present the examples of United States and New Zeeland? Please correct the sentence as it is difficult for the reader to read.

Reply: Thank you very much for your comment. The original purpose is to show the burden of the diseases. We have removed the examples in New Zealand in the revised manuscript.

  1. 1.1: This is not an introduction that describes Stem-cell based therapy.

Reply: Thank you very much for your comments. We have revised introduction.

It is difficult to continue peer review due to many careless mistakes and fatal errors in logical composition.

Reply: We are sorry for the careless mistakes. We have extensively revised the manuscript. We are grateful for your comments to improve the quality of the manuscript.

Reviewer 2 Report

General Comments

This manuscript provides a guideline of what is required for the application of EVs as a pharmaceutical agent to promote tissue repair.

The assumption is that EVs produce by a single cell are identical in that they have a similar, if not identical, contents of DNA, RNA, proteins, etc.  But, is this true?  What does the literature say regarding this matter.  Also, how much variability is there in EVs produced by a given population of cells cultured under identical conditions?  Does this change with passage number?

l. 418; As MSC-EVs carry inherently complex cell-specific cargos of proteins, lipids and genetic materials, standardized characterization is challenging.  What is the evidence is there for homogeneity or lack thereof for a single batch of EVs?  This needs to be included.

There are some minor issues with grammar.

Specific Comments

l. 70-71; EVs can be sterilized by filtration -- Filtration does not remove virus particles.

l. 75-76 the intrinsic ability to cross the epithelium and blood-brain barrier and hence may be

useful for the delivery of drugs   This should be referenced.

Author Response

The assumption is that EVs produce by a single cell are identical in that they have a similar, if not identical, contents of DNA, RNA, proteins, etc.  But, is this true?  What does the literature say regarding this matter.  Also, how much variability is there in EVs produced by a given population of cells cultured under identical conditions?  Does this change with passage number?

Reply: The content of EVs is affected by the heterogenicity of the donor MSCs. The phenotypes and biological activities of MSCs depend on their origin (biological niche) or the conditions of potential donors (age, diseases, obesity or unknown factors) (Costa et al., 2021). The artificial microenvironment of MSCs such as O2 tension, substrate and extracellular matrix cues, culture media, inflammatory stimuli or genetic manipulations can also influence their resulting phenotypes and hence paracrine activities (Costa et al., 2021). Exosomes derived from umbilical cord derived MSCs cultured in two different cell culture media showed different surface composition and cytokine content (Kim et al., 2021). The passage number of MSCs also influences the biological activities of EVs. Exosomes isolated from early passage of rat BMSCs exhibited higher neuroprotective potential compared to exosomes derived from later passages of BMSCs (Venugopal et al., 2017).

The mechanisms of biogenesis of EVs also influence their content. At least three major subpopulations of EVs produced by different mechanisms, namely exosomes, microvesicles and apoptotic bodies, are reported. Immortalized E1-MYC 16.3 human embryonic stem cells-derived MSCs were reported to produce at least 3 distinct 100 nm EV types by different biogenesis pathways that could be distinguished by their membrane lipid composition, proteome and RNA cargos (Lai et al., 2016). In one study, BMSC-EVs were fractionated into different density fractions (Collino et al., 2017). The different EV gradient fractions were heterogeneous in quantity and expression of classical exosomal markers. The miRNA and protein profiles of these EV fractions were different, and they also showed differential effects on renal tubular cells in terms of degree of internationalization, stimulation of cell proliferation and inhibition of apoptosis. Therefore, the content of MSC-EVs is heterogeneous with specific signatures accounting for the biological activity of different EV subpopulations and EVs produced by a given population of cells cultured under identical conditions are not identical.

The above information is added in the revised manuscript.

References:

Costa, L. A., Eiro, N., Fraile, M., Gonzalez, L. O., Saá, J., Garcia-Portabella, P., Vega, B., Schneider, J., Vizoso, F. J. Functional heterogeneity of mesenchymal stem cells from natural niches to culture conditions: implications for further clinical uses. Cell Mol Life Sci. 202178(2), 447–467.

Kim, J. Y., Rhim, W. K., Seo, H. J., Lee, J. Y., Park, C. G., Han, D. K. Comparative Analysis of MSC-Derived Exosomes Depending on Cell Culture Media for Regenerative Bioactivity. Tissue Eng Regen Med. 2021,18(3), 355–367.

Venugopal, C., Shamir, C., Senthilkumar, S., Babu, J. V., Sonu, P. K., Nishtha, K. J., Rai, K. S., K, S., Dhanushkodi, A. Dosage and Passage Dependent Neuroprotective Effects of Exosomes Derived from Rat Bone Marrow Mesenchymal Stem Cells: An In Vitro Analysis. Curr Gene Ther. 2017, 17(5), 379–390.

Lai, R. C., Tan, S. S., Yeo, R. W., Choo, A. B., Reiner, A. T., Su, Y., Shen, Y., Fu, Z., Alexander, L., Sze, S. K., Lim, S. K. MSC secretes at least 3 EV types each with a unique permutation of membrane lipid, protein and RNA. J Extracell Vesicles 20165, 29828. 

Collino, F., Pomatto, M., Bruno, S., Lindoso, R. S., Tapparo, M., Sicheng, W., Quesenberry, P., Camussi, G. Exosome and Microvesicle-Enriched Fractions Isolated from Mesenchymal Stem Cells by Gradient Separation Showed Different Molecular Signatures and Functions on Renal Tubular Epithelial Cells. Stem Cell Rev Rep. 201713(2), 226–243.

  1. 418; As MSC-EVs carry inherently complex cell-specific cargos of proteins, lipids and genetic materials, standardized characterization is challenging. What is the evidence is there for homogeneity or lack thereof for a single batch of EVs? This needs to be included.

Reply: As discussed above, the composition of MSC-EVs is heterogenous even under controlled conditions due to different biogenesis pathways. While fractionation or sorting may help to identify more homogeneous subpopulations of EVs, careful control of the donor cell source, cell culture conditions and EV enrichment process is crucial for clinical translation of EV-based therapies, as they standardize the EV composition and content.

The above information is added in the revised manuscript.

There are some minor issues with grammar.

Reply: We are sorry for the grammatical mistakes. We have checked the manuscript carefully and corrected all the grammatical mistakes.

Specific Comments

  1. 70-71; EVs can be sterilized by filtration -- Filtration does not remove virus particles.

Reply: The reviewer is right that viral particles cannot be removed by filtration as the they are of similar size as EVs. The manuscript is revised.

  1. 75-76 the intrinsic ability to cross the epithelium and blood-brain barrier and hence may be useful for the delivery of drugs This should be referenced.

Reply: The following references are added in the revised manuscript.

Wood, M. J., O'Loughlin, A. J., Samira, L. Exosomes and the blood-brain barrier: implications for neurological diseases. Ther Deliv. 2011, 2(9), 1095–1099.

Skog, J., Wurdinger, T., van Rijn, S., Meijer, D. H., Gainche, L., Sena-Esteves, M., Curry, W. T. Jr., Carter, B. S., Krichevsky, A. M., Breakefield, X. O. Glioblastoma microvesicles transport RNA and proteins that promote tumour growth and provide diagnostic biomarkers. Nat Cell Biol. 2008, 10, 1470–6. 

Alvarez-Erviti, L., Seow, Y., Yin, H., Betts, C., Lakhal, S., Wood, M. J. Delivery of siRNA to the mouse brain by systemic injection of targeted exosomes. Nat Biotechnol. 2011, 29, 341–5.

Saint-Pol, J., Gosselet, F., Duban-Deweer, S., Pottiez, G., Karamanos, Y. Targeting and Crossing the Blood-Brain Barrier with Extracellular Vesicles. Cells 20209(4), 851.

Reviewer 3 Report

This manuscript discusses recent advances of extracellular vesicles (EVs) derived from mesenchymal stromal cells (MSCs) for the treatment of tendon and ligament injuries. In addition, practical issues are mentioned for successful clinical application of MSC-derived EV-based products for the treatment of tendon and ligament injuries. The review focuses on Good Manufacturing Practice (GMP)-compliant manufacturing and quality control (parent cell source, culture conditions, concentration method, quantity, identity, purity, and impurities, sterility, potency, reproducibility, storage and formulation), safety and efficacy issues for clinical translation of MSC EV-based therapies. In general this is a thorough review of the GMP requirements for EV production. Some concerns needs to be addressed below.  

1. The reason to focus discussions of EVs for tendon and ligament repair in Section 1 and not other diseases or conditions is not apparent. An improved rationale or transition to the rest of the review needs to be provided if tendon/ligament repair is to be introduced in Section 1. The authors should consider discussing the requirements of GMP of EVs for tendon/ligament repair from section 2 onwards.

2. In the introduction, discuss current understandings of how MSC-EVs promote tendon/ligament repair i.e. the active molecule(s) present in EVs that contribute to repair.  

3. Indicate the references for Tables 3 and 4.

4. Check typos and grammatical errors e.g. “Despites’ (line 605).

5. The review mentioned advantages of EVs as therapeutics for tendon and ligament repair. Also indicate their limitations as therapeutics, if any.   

Author Response

  1. The reason to focus discussions of EVs for tendon and ligament repair in Section 1 and not other diseases or conditions is not apparent. An improved rationale or transition to the rest of the review needs to be provided if tendon/ligament repair is to be introduced in Section 1. The authors should consider discussing the requirements of GMP of EVs for tendon/ligament repair from section 2 onwards.

Reply: Thank you very much for your comment. We have revised the manuscript by putting the discussion of tendon/ligament repair in section 2.

  1. In the introduction, discuss current understandings of how MSC-EVs promote tendon/ligament repair i.e. the active molecule(s) present in EVs that contribute to repair.

Reply: The administration of MSC-EVs was generally found to promote cell proliferation and migration (Gissi et al., 2020; Zhang et al., 2020; Yu et al., 2020; Liu et al., 2021a; Song et al., 2022; Fu et al., 2021; Ren et al., 2021; Shi et al., 2020; Liu et al., 2021b; Zhu et al., 2022), suppress tissue inflammation and apoptosis (Yao et al., 2020; Zhang et al., 2020; Shi et al., 2019; Liu et al., 2021a; Huang et al., 2020; Shi et al., 2020; Liu et al., 2021b; Zhu et al., 2022), modulate inflammatory response of macrophages (Chamberlain et al., 21019; Shen et al., 2020; Shi et al., 2019; Liu et al., 2021a; Huang et al., 2020; Shi et al., 2020), increase collagen deposition (Gissi et al., 2020; Yao et al., 2021; Song et al., 2022;), reduce fatty infiltration (Wang et al., 2020) and promote angiogenesis (Huang et al., 2020) during tendon and ligament repair.

The active molecules in MSC-EVs that contribute to tendon and ligament repair are not entirely known. However, numerous studies have shown that MSC-EVs are rich in miRNA, which contribute to tendon and ligament repair. miRNA sequencing showed that human umbilical cord MSC-derived exosomes (HUMSC-Exos) expressed an antagonist to miR-21a-3p as miR-21a-3p was under expressed in the exosomes compared to the parent cells. The inhibition of exosomal miR-21a-3p in HUMSC-Exos inactivated RelA/p65 (a core element in the NF-kB pathway involved in inflammation and fibrosis) and reduced the protein expression of Cox2 and α-SMA in rat fibroblasts (Yao et al., 2020). Besides, endoplasmic reticulum stress (ERS)-associated proteins (GRP78, CHOP) and pro-apoptotic protein (Bax) were presented in EVs derived from HUMSCs which might explain their anti-adhesive effect on traumatic tendon injury (Li et al., 2020). In addition, HUMSC-Exos promoted tendon healing via miR-27b-3p-mediated suppression of ARHGAP5, resulting in RhoA activation, increased proliferation and migration of primary injured tenocytes (Han et al., 2022). In another study, rat EVs derived from bone marrow-derived stromal cells (BMSC-EVs) were reported to express pro-collagen1A2 and MMP14 proteins, which are important factors for tendon extracellular matrix remodeling. Pro-collagen1A2 was expressed on the membrane surface of BMSC-EVs. Pretreatment of BMSC-EVs with trypsin abrogated their effects on tendon cell proliferation and migration, and the expression of collagen type I, suggesting that the biological effects of EVs depended on the interaction of membrane-bound proteins with the recipient tendon cells (Gissi et al., 2020). Besides, miRNA sequencing indicated a significant higher level of miR-29a-3p in HUMSC-Exos compared to HUMSCs (Yao et al., 2021). The level of miR-29a-3p in HUMSC-Exos-treated Achilles tendons was also significantly elevated and HUMSC-Exos overexpressing miR-29a-3p amplified the effects of HUMSC-Exos on tendon healing in vivo (Yao et al., 2021). In addition, exosomes derived from TDSCs (TDSC-Exos) contained miR-144-3p and enhanced tendon repair through miR-144-3p-regulated tenocyte proliferation and migration (Song et al., 2022). Furthermore, exosomes derived from Scx overexpressing PDGFRα(+) BMSCs reduced osteoclastogenesis and improved tendon-bone healing strength via exosomal miR-6924-5p (Feng et al., 2021).

References:

Chamberlain, C. S., Clements, A., Kink, J. A., Choi, U., Baer, G. S., Halanski, M. A., Hematti, P., Vanderby, R. Extracellular Vesicle-Educated Macrophages Promote Early Achilles Tendon Healing. Stem Cells 2019, 37(5), 652–662.

Shen, H., Yoneda, S., Abu-Amer, Y., Guilak, F., Gelberman, R. H. Stem cell-derived extracellular vesicles attenuate the early inflammatory response after tendon injury and repair. J Orthop Res. 2020, 38(1), 117–127.

Yao, Z., Li, J., Wang, X., Peng, S., Ning, J., Qian, Y., Fan, C. MicroRNA-21-3p Engineered Umbilical Cord Stem Cell-Derived Exosomes Inhibit Tendon Adhesion. J Inflamm Res. 2020, 13, 303–316.

Li, J., Yao, Z., Xiong, H., Cui, H., Wang, X., Zheng, W., Qian, Y., Fan, C. Extracellular vesicles from hydroxycamptothecin primed umbilical cord stem cells enhance anti-adhesion potential for treatment of tendon injury. Stem Cell Res Ther. 2020,11(1), 500.

Han, Q., Wang, S., Chen, D., Gan, D., Wang, T. Exosomes derived from human umbilical cord mesenchymal stem cells reduce tendon injuries via the miR-27b-3p/ARHGAP5/RhoA signaling pathway. Acta Biochim Biophys Sin (Shanghai) 2022, 54(2), 232–242. 

Gissi, C., Radeghieri, A., Antonetti Lamorgese Passeri, C., Gallorini, M., Calciano, L., Oliva, F., Veronesi, F., Zendrini, A., Cataldi, A., Bergese, P., Maffulli, N., Berardi, A. C. Extracellular vesicles from rat-bone-marrow mesenchymal stromal/stem cells improve tendon repair in rat Achilles tendon injury model in dose-dependent manner: A pilot study. PloS One 2020, 15(3), e0229914.

Zhang, M., Liu, H., Cui, Q., Han, P., Yang, S., Shi, M., Zhang, T., Zhang, Z., Li, Z. Tendon stem cell-derived exosomes regulate inflammation and promote the high-quality healing of injured tendon. Stem Cell Res Ther. 202011(1), 402.

Yao, Z., Li, J., Xiong, H., Cui, H., Ning, J., Wang, S., Ouyang, X., Qian, Y., Fan, C. MicroRNA engineered umbilical cord stem cell-derived exosomes direct tendon regeneration by mTOR signaling. J Nanobiotechnology 2021, 19(1), 169.

Shi, Z., Wang, Q., Jiang, D. Extracellular vesicles from bone marrow-derived multipotent mesenchymal stromal cells regulate inflammation and enhance tendon healing. J Transl Med2019, 17(1), 211.

Yu, H., Cheng, J., Shi, W., Ren, B., Zhao, F., Shi, Y., Yang, P., Duan, X., Zhang, J., Fu, X., Hu, X., Ao, Y. Bone marrow mesenchymal stem cell-derived exosomes promote tendon regeneration by facilitating the proliferation and migration of endogenous tendon stem/progenitor cells. Acta Biomater. 2020106, 328–341.

Liu, H., Zhang, M., Shi, M., Zhang, T., Lu, W., Yang, S., Cui, Q., Li, Z. Adipose-derived mesenchymal stromal cell-derived exosomes promote tendon healing by activating both SMAD1/5/9 and SMAD2/3. Stem Cell Res Ther. 2021, 12(1), 338.

Song, K., Jiang, T., Pan, P., Yao, Y., Jiang, Q. Exosomes from tendon derived stem cells promote tendon repair through miR-144-3p-regulated tenocyte proliferation and migration. Stem Cell Res Ther. 202213(1), 80. 

Wang, C., Hu, Q., Song, W., Yu, W., He, Y. Adipose Stem Cell-Derived Exosomes Decrease Fatty Infiltration and Enhance Rotator Cuff Healing in a Rabbit Model of Chronic Tears. Am J Sports Med. 202048(6), 1456–1464.

Han, L., Liu, H., Fu, H., Hu, Y., Fang, W., Liu, J. Exosome-delivered BMP-2 and polyaspartic acid promotes tendon bone healing in rotator cuff tear via Smad/RUNX2 signaling pathway. Bioengineered 2022,13(1), 1459–1475.

Huang, Y., He, B., Wang, L., Yuan, B., Shu, H., Zhang, F., Sun, L. Bone marrow mesenchymal stem cell-derived exosomes promote rotator cuff tendon-bone healing by promoting angiogenesis and regulating M1 macrophages in rats. Stem Cell Res Ther. 2020, 11(1), 496.

Fu, G., Lu, L., Pan, Z., Fan, A., Yin, F. (2021). Adipose-derived stem cell exosomes facilitate rotator cuff repair by mediating tendon-derived stem cells. Regen Med. 2021,16(4), 359–372.

Ren, Y., Zhang, S., Wang, Y., Jacobson, D. S., Reisdorf, R. L., Kuroiwa, T., Behfar, A., Moran, S. L., Steinmann, S. P., Zhao, C. Effects of purified exosome product on rotator cuff tendon-bone healing in vitro and in vivo. Biomaterials 2021276, 121019.

Shi, Y., Kang, X., Wang, Y., Bian, X., He, G., Zhou, M., Tang, K. Exosomes Derived from Bone Marrow Stromal Cells (BMSCs) Enhance Tendon-Bone Healing by Regulating Macrophage Polarization. Med Sci Monit. 2020, 26, e923328.

Feng, W., Jin, Q., Ming-Yu, Y., Yang, H., Xu, T., You-Xing, S., Xu-Ting, B., Wan, C., Yun-Jiao, W., Huan, W., Ai-Ning, Y., Yan, L., Hong, T., Pan, H., Mi-Duo, M., Gang, H., Mei, Z., Xia, K., Kang-Lai, T. MiR-6924-5p-rich exosomes derived from genetically modified Scleraxis-overexpressing PDGFRα(+) BMMSCs as novel nanotherapeutics for treating osteolysis during tendon-bone healing and improving healing strength. Biomaterials2021, 279, 121242.

Wang, Y., He, G., Guo, Y., Tang, H., Shi, Y., Bian, X., Zhu, M., Kang, X., Zhou, M., Lyu, J., Yang, M., Mu, M., Lai, F., Lu, K., Chen, W., Zhou, B., Zhang, J., Tang, K. Exosomes from tendon stem cells promote injury tendon healing through balancing synthesis and degradation of the tendon extracellular matrix. J Cell Mol Med. 2019, 23(8), 5475–5485.

Liu, A., Wang, Q., Zhao, Z., Wu, R., Wang, M., Li, J., Sun, K., Sun, Z., Lv, Z., Xu, J., Jiang, H., Wan, M., Shi, D., Mao, C. Nitric Oxide Nanomotor Driving Exosomes-Loaded Microneedles for Achilles Tendinopathy Healing. ACS Nano. 2021, 10.1021/acsnano.1c03177. 

Zhu, Z., Gao, R., Ye, T., Feng, K., Zhang, J., Chen, Y., Xie, Z., Wang, Y. The Therapeutic Effect of iMSC-Derived Small Extracellular Vesicles on Tendinopathy Related Pain Through Alleviating Inflammation: An in vivo and in vitro Study. J Inflamm Res. 202215, 1421–1436. 

Chamberlain, C. S., Kink, J. A., Wildenauer, L. A., McCaughey, M., Henry, K., Spiker, A. M., Halanski, M. A., Hematti, P., Vanderby, R. Exosome-educated macrophages and exosomes differentially improve ligament healing. Stem Cells 2021, 39(1), 55–61. 

Kornicka-Garbowska, K., PÄ™dziwiatr, R., Woźniak, P., Kucharczyk, K., Marycz, K. Microvesicles isolated from 5-azacytidine-and-resveratrol-treated mesenchymal stem cells for the treatment of suspensory ligament injury in horse-a case report. Stem Cell Res Ther, 201910(1), 394. 

  1. Indicate the references for Tables 3 and 4.

Reply: Table 3 and 4 are based on our understanding and synthesis of the literature after reading many papers. Some of these papers are already cited in the review as we discuss each topic in text. The key references for each table are now included.

Table 3

Poupardin, R.; Wolf, M.; Strunk, D. Adherence to minimal experimental requirements for defining extracellular vesicles and their functions. Adv Drug Deliv Rev2021, 176, 113872.

Witwer, K. W.; Van Balkom, B.; Bruno, S.; Choo, A.; Dominici, M.; Gimona, M.; Hill, A. F.; De Kleijn, D.; Koh, M.; Lai, R. C.; Mitsialis, S. A.; Ortiz, L. A.; Rohde, E.; Asada, T.; Toh, W. S.; Weiss, D. J.; Zheng, L.; Giebel, B.; Lim, S. K. Defining mesenchymal stromal cell (MSC)-derived small extracellular vesicles for therapeutic applications. J Extracell Vesicles2019, 8(1), 1609206.

Andriolo, G.; Provasi, E.; Brambilla, A.; Lo Cicero, V.; Soncin, S.; Barile, L.; Turchetto, L.; Radrizzani, M. GMP-Grade Methods for Cardiac Progenitor Cells: Cell Bank Production and Quality Control. Methods Mol Biol. 2021, 2286, 131–166.

Lechanteur, C.; Briquet, A.; Bettonville, V.; Baudoux, E.; Beguin, Y. MSC Manufacturing for Academic Clinical Trials: From a Clinical-Grade to a Full GMP-Compliant Process. Cells 2021, 10(6), 1320.

Guadix, J. A.; López-Beas, J.; Clares, B.; Soriano-Ruiz, J. L.; Zugaza, J. L.; Gálvez-Martín, P. Principal Criteria for Evaluating the Quality, Safety and Efficacy of hMSC-Based Products in Clinical Practice: Current Approaches and Challenges. Pharmaceutics 2019, 11(11), 552.

Robb, K. P.; Fitzgerald, J. C.; Barry, F.; Viswanathan, S. Mesenchymal stromal cell therapy: progress in manufacturing and assessments of potency. Cytotherapy 2019, 21(3), 289–306.

Dominici, M.; Le Blanc, K.; Mueller, I.; Slaper-Cortenbach, I.; Marini, F.; Krause, D.; et al. Minimal criteria for defining multipotent mesenchymal stromal cells. The International Society for Cellular Therapy position statement. Cytotherapy 2006, 8, 315-317

Council of Europe. Nucleated cell count and viability. In The European Pharmacopoeia, 9th ed.; EDQM: Strasbourg, France, 2017, 2.7.29

Seaver, S. A new United States Pharmacopeia (USP) Chapter 1046: cell and gene therapy products. Cytotherapy 2000, 2(1), 45-49.

Table 4

Poupardin, R.; Wolf, M.; Strunk, D. Adherence to minimal experimental requirements for defining extracellular vesicles and their functions. Adv Drug Deliv Rev2021, 176, 113872.

Lener, T.; Gimona, M.; Aigner, L.; Börger, V.; Buzas, E.; Camussi, G.; Chaput, N.; Chatterjee, D.; Court, F. A.; Del Portillo, H. A.; O'Driscoll, L.; Fais, S.; Falcon-Perez, J. M.; Felderhoff-Mueser, U.; Fraile, L.; Gho, Y. S.; Görgens, A.; Gupta, R. C.; Hendrix, A.; Hermann, D. M.; … Giebel, B. Applying extracellular vesicles based therapeutics in clinical trials - an ISEV position paper. J Extracell Vesicles2015, 4, 30087. 

Witwer, K. W.; Van Balkom, B.; Bruno, S.; Choo, A.; Dominici, M.; Gimona, M.; Hill, A. F.; De Kleijn, D.; Koh, M.; Lai, R. C.; Mitsialis, S. A.; Ortiz, L. A.; Rohde, E.; Asada, T.; Toh, W. S.; Weiss, D. J.; Zheng, L.; Giebel, B.; Lim, S. K. Defining mesenchymal stromal cell (MSC)-derived small extracellular vesicles for therapeutic applications. J Extracell Vesicles2019, 8(1), 1609206.

Silva, A.; Morille, M.; Piffoux, M.; Arumugam, S.; Mauduit, P.; Larghero, J.; Bianchi, A.; Aubertin, K.; Blanc-Brude, O.; Noël, D.; Velot, E.; Ravel, C.; Elie-Caille, C.; Sebbagh, A.; Boulanger, C.; Wilhelm, C.; Rahmi, G.; Raymond-Letron, I.; Cherukula, K.; Montier, T.; … Banzet, S. Development of extracellular vesicle-based medicinal products: A position paper of the group "Extracellular Vesicle translatiOn to clinicaL perspectiVEs - EVOLVE France". Adv. Drug Deliv Rev. 2021, 179, 114001.

Chen, Y. S.; Lin, E. Y.; Chiou, T. W.; Harn, H. J. Exosomes in clinical trial and their production in compliance with good manufacturing practice. Ci Ji Yi Xue Za Zhi 2019, 32(2), 113–120.

Osteikoetxea, X.; Balogh, A.; Szabó-Taylor, K.; Németh, A.; Szabó, T. G.; Pálóczi, K.; Sódar, B.; Kittel, Á.; György, B.; Pállinger, É.; Matkó, J.; Buzás, E. I. Improved characterization of EV preparations based on protein to lipid ratio and lipid properties. PLoS One 2015, 10(3), e0121184.

Webber, J.; Clayton, A. How pure are your vesicles? J Extracell Vesicles 2013, 10, 2.

Maiolo, D.; Paolini, L.; Di Noto, G.; Zendrini, A.; Berti, D.; Bergese, P.; Ricotta, D. Colorimetric nanoplasmonic assay to determine purity and titrate extracellular vesicles. Anal Chem. 2015, 87(8), 4168-76.

Mihály, J.; Deák, R.; Szigyártó, I. C.; Bóta, A.; Beke-Somfai, T.; Varga, Z. Characterization of extracellular vesicles by IR spectroscopy: Fast and simple classification based on amide and CH stretching vibrations. Biochim Biophys Acta Biomembr. 2017, 1859(3), 459-466.

Veerman, R. E.; Teeuwen, L.; Czarnewski, P.; Güclüler Akpinar, G.; Sandberg, A.; Cao, X.; Pernemalm, M.; Orre, L. M.; Gabrielsson, S.; Eldh, M. Molecular evaluation of five different isolation methods for extracellular vesicles reveals different clinical applicability and subcellular origin. J Extracell Vesicles 2021, 10(9), e12128.

Rohde, E.; Pachler, K.; Gimona, M. Manufacturing and characterization of extracellular vesicles from umbilical cord-derived mesenchymal stromal cells for clinical testing. Cytotherapy 2019, 21(6), 581–592.

Buschmann, D., Mussack, V., Byrd, J. B. Separation, characterization, and standardization of extracellular vesicles for drug delivery applications. Adv Drug Deliv Rev. 2021, 174, 348–368.

Gimona, M.; Pachler, K.; Laner-Plamberger, S.; Schallmoser, K.; Rohde, E. Manufacturing of Human Extracellular Vesicle-Based Therapeutics for Clinical Use. Int J Mol Sci. 2017, 18(6), 1190.

Théry, C.; Witwer, K. W.; Aikawa, E.; Alcaraz, M. J.; Anderson, J. D.; Andriantsitohaina, R.; Antoniou, A.; Arab, T.; Archer, F.; Atkin-Smith, G. K.; Ayre, D. C.; Bach, J. M.; Bachurski, D.; Baharvand, H.; Balaj, L.; Baldacchino, S.; Bauer, N. N.; Baxter, A. A.; Bebawy, M.; Beckham, C.; Bedina Zavec, A.; Benmoussa, A.; Berardi, A. C.; Bergese, P.; Bielska, E.; Blenkiron, C.; Bobis-Wozowicz, S.; Boilard, E.; Boireau, W.; Bongiovanni, A.; Borràs, F. E.; Bosch, S.; Boulanger, C. M.; Breakefield, X.; Breglio, A. M.; Brennan, M. Á.; Brigstock, D. R.; Brisson, A.; Broekman, M. L.; Bromberg, J. F.; Bryl-Górecka, P.; Buch, S.; Buck, A. H.; Burger, D.; Busatto, S.; Buschmann, D.; Bussolati, B.; Buzás, E. I.; Byrd, J. B.; Camussi, G.; Carter, D. R.; Caruso, S.; Chamley, L. W.; Chang, Y. T.; Chen, C.; Chen, S.; Cheng, L.; Chin, A. R.; Clayton, A.; Clerici, S. P.; Cocks, A.; Cocucci, E.; Coffey, R. J.; Cordeiro-da-Silva, A.; Couch, Y.; Coumans, F. A.; Coyle, B.; Crescitelli, R.; Criado, M. F.; D'Souza-Schorey, C.; Das, S.; Datta Chaudhuri, A.; de Candia, P.; De Santana, E. F.; De Wever, O.; Del Portillo, H. A.; Demaret, T.; Deville, S.; Devitt, A.; Dhondt, B.; Di Vizio, D.; Dieterich, L. C.; Dolo, V.; Dominguez Rubio, A. P.; Dominici, M.; Dourado, M. R.; Driedonks, T. A.; Duarte, F. V.; Duncan, H. M.; Eichenberger, R. M.; Ekström, K.; El Andaloussi, S.; Elie-Caille, C.; Erdbrügger, U.; Falcón-Pérez, J. M.; Fatima, F.; Fish, J. E.; Flores-Bellver, M.; Försönits, A.; Frelet-Barrand, A.; Fricke, F.; Fuhrmann, G.; Gabrielsson, S.; Gámez-Valero, A.; Gardiner, C.; Gärtner, K.; Gaudin, R.; Gho, Y. S.; Giebel, B.; Gilbert, C.; Gimona, M.; Giusti, I.; Goberdhan, D. C.; Görgens, A.; Gorski, S. M.; Greening, D. W.; Gross, J. C.; Gualerzi, A.; Gupta, G. N.; Gustafson, D.; Handberg, A.; Haraszti, R. A.; Harrison, P.; Hegyesi, H.; Hendrix, A.; Hill, A. F.; Hochberg, F. H.; Hoffmann, K. F.; Holder, B.; Holthofer, H.; Hosseinkhani, B.; Hu, G.; Huang, Y.; Huber, V.; Hunt S.; Ibrahim AG.; Ikezu T.; Inal JM.; Isin M.; Ivanova A.; Jackson HK.; Jacobsen S.; Jay SM.; Jayachandran, M.; Jenster, G.; Jiang, L.; Johnson, S. M.; Jones, J. C.; Jong, A.; Jovanovic-Talisman, T.; Jung, S.; Kalluri, R.; Kano, S. I.; Kaur, S.; Kawamura, Y.; Keller, E. T.; Khamari, D.; Khomyakova, E.; Khvorova, A.; Kierulf, P.; Kim, K. P.; Kislinger, T.; Klingeborn, M.; Klinke, D. J. 2nd.; Kornek, M.; Kosanović, M. M.; Kovács, Á. F.; Krämer-Albers, E. M.; Krasemann, S.; Krause, M.; Kurochkin, I. V.; Kusuma, G. D.; Kuypers, S.; Laitinen, S.; Langevin, S. M.; Languino, L. R.; Lannigan, J.; Lässer, C.; Laurent, L. C.; Lavieu, G.; Lázaro-Ibáñez, E.; Le Lay, S.; Lee, M. S.; Lee, Y. X, F.; Lemos, D. S.; Lenassi, M.; Leszczynska, A.; Li, I. T.; Liao, K.; Libregts, S. F.; Ligeti, E.; Lim, R.; Lim, S. K.; LinÄ“, A.; Linnemannstöns, K.; Llorente, A.; Lombard, C. A.; Lorenowicz, M. J.; Lörincz, Á. M.; Lötvall, J.; Lovett, J.; Lowry, M. C.; Loyer, X.; Lu, Q.; Lukomska, B.; Lunavat, T. R.; Maas, S. L.; Malhi, H.; Marcilla, A.; Mariani, J.; Mariscal, J.; Martens-Uzunova, E. S.; Martin-Jaular, L.; Martinez, M. C.; Martins, V. R.; Mathieu, M.; Mathivanan, S.; Maugeri, M.; McGinnis, L. K.; McVey, M. J.; Meckes, D. G. Jr.; Meehan, K. L.; Mertens, I.; Minciacchi, V. R.; Möller, A.; Møller Jørgensen, M.; Morales-Kastresana, A.; Morhayim, J.; Mullier, F.; Muraca, M.; Musante, L.; Mussack, V.; Muth, D. C.; Myburgh, K. H.; Najrana, T.; Nawaz, M.; Nazarenko, I.; Nejsum, P.; Neri, C.; Neri, T.; Nieuwland, R.; Nimrichter, L.; Nolan, J. P.; Nolte-'t Hoen, E. N.; Noren Hooten, N.; O'Driscoll, L.; O'Grady, T.; O'Loghlen, A.; Ochiya, T.; Olivier, M.; Ortiz, A.; Ortiz, L. A.; Osteikoetxea, X.; Østergaard, O.; Ostrowski, M.; Park, J.; Pegtel, D. M.; Peinado, H.; Perut, F.; Pfaffl, M. W.; Phinney, D. G.; Pieters, B. C.; Pink, R. C.; Pisetsky, D. S.; Pogge von Strandmann, E.; Polakovicova, I.; Poon, I. K.; Powell, B. H.; Prada, I.; Pulliam, L.; Quesenberry, P.; Radeghieri, A.; Raffai, R. L.; Raimondo, S.; Rak, J.; Ramirez, M. I.; Raposo, G.; Rayyan, M. S.; Regev-Rudzki, N.; Ricklefs, F. L.; Robbins, P. D.; Roberts, D. D.; Rodrigues, S. C.; Rohde, E.; Rome, S.; Rouschop, K. M.; Rughetti, A.; Russell, A. E.; Saá, P.; Sahoo, S.; Salas-Huenuleo, E.; Sánchez, C.; Saugstad, J. A.; Saul, M. J.; Schiffelers, R. M.; Schneider, R.; Schøyen, T. H.; Scott, A.; Shahaj, E.; Sharma, S.; Shatnyeva, O.; Shekari, F.; Shelke, G. V.; Shetty, A. K.; Shiba, K.; Siljander, P. R.; Silva, A. M.; Skowronek, A.; Snyder, O. L. 2nd.; Soares, R. P.; Sódar, B. W.; Soekmadji, C.; Sotillo, J.; Stahl, P. D.; Stoorvogel, W.; Stott, S. L.; Strasser, E. F.; Swift, S.; Tahara, H.; Tewari, M.; Timms, K.; Tiwari, S.; Tixeira, R.; Tkach, M.; Toh, W. S.; Tomasini, R.; Torrecilhas, A. C.; Tosar, J. P.; Toxavidis, V.; Urbanelli, L.; Vader, P.; van Balkom, B. W.; van der Grein, S. G.; Van Deun, J.; van Herwijnen, M. J.; Van Keuren-Jensen, K.; van Niel, G.; van Royen, M. E.; van Wijnen, A. J.; Vasconcelos, M. H.; Vechetti, I. J. Jr.; Veit, T. D.; Vella, L. J.; Velot, É.; Verweij, F. J.; Vestad, B.; Viñas, J. L.; Visnovitz, T.; Vukman, K. V.; Wahlgren, J.; Watson, D. C.; Wauben, M. H.; Weaver, A.; Webber, J. P.; Weber, V.; Wehman, A. M.; Weiss, D. J.; Welsh, J. A.; Wendt, S.; Wheelock, A. M.; Wiener, Z.; Witte, L.; Wolfram, J.; Xagorari, A.; Xander, P.; Xu, J.; Yan, X.; Yáñez-Mó, M.; Yin, H.; Yuana, Y.; Zappulli, V.; Zarubova, J.; Žėkas, V.; Zhang, J. Y.; Zhao, Z.; Zheng, L.; Zheutlin, A. R.; Zickler, A. M.; Zimmermann, P.; Zivkovic, A. M.; Zocco, D.; Zuba-Surma, E. K. Minimal information for studies of extracellular vesicles 2018 (MISEV2018): a position statement of the International Society for Extracellular Vesicles and update of the MISEV2014 guidelines. J Extracell Vesicles 2018, 7(1), 1535750.

Bari, E.; Perteghella, S.; Catenacci, L.; Sorlini, M.; Croce, S.; Mantelli, M.; Avanzini, M. A.; Sorrenti, M.; Torre, M. L. Freeze-dried and GMP-compliant pharmaceuticals containing exosomes for acellular mesenchymal stromal cell immunomodulant therapy. Nanomedicine (Lond). 2019, 14(6), 753-765.

  1. Check typos and grammatical errors e.g. “Despites’ (line 605).

Reply: we are sorry for the typos. It is corrected.

  1. The review mentioned advantages of EVs as therapeutics for tendon and ligament repair. Also indicate their limitations as therapeutics, if any.

Reply: Thank you very much for your comments. The following paragraph is added in the last section of the revised manuscript.

There are technical challenges for clinical application of MSC-EVs. First, the biochemical composition of MSC-EVs remains unclear. The production or uptake mechanisms are poorly described. GMP standards for clinical grade production, storage and recovery are therefore lacking. Second, the drug loading efficiency of EVs is relatively lower than that for liposomes. Third, engineered MSC-EVs containing transgene products are categorized as ATPs and hence need to meet more stringent regulatory requirements. Fourth, as the half-life of MSC-EVs is short and they cannot replicate themselves like MSCs, large amount of MSC-EVs is required for clinical application. Large scale and efficient production of MSC-EVs remain difficult at present. Finally, the physiochemical properties, particle size and concentration as well as content of MSC-EVs have been the focus of investigation. The MoA or potency of MSC-EVs remains relatively unexplored, resulting in the lack of appropriate functional assays. To overcome the bottleneck for MSC-EV-based therapies, clinical-grade MSCs meeting requirements for therapeutic use and transplantation are needed for MSC-EV production. Conditions to ensure sustainable release of MSC-EVs should be established. The MSC-EV specification varies with the enrichment methods. Therefore, it is important to choose a cost-effective, scalable and automatable concentration method as early as possible to avoid changing the method and hence MSC-EV specification due to scaling up of production. While there is some consensus of using -80 oC for MSC-EV transportation and storage, storage at -80 oC poses challenges in transportation and is a cost-ineffective approach. Alternative methods such as lyophilization may improve MSC-EV stability during storage and transportation. The route of administration greatly influences the therapeutic efficacy and safety of MSC-EVs and should be determined prior to clinical application. Effort to develop high-throughput and precise quantification methods for assessing MoA or potency will greatly facilitate clinical translation of MSC-EV-based therapies.”

Reviewer 4 Report

This is a very detailed review for the translation of mesenchymal stromal cells derived EVs for the treatment of tendon and ligament injuries. The manuscript can be accepted after one minor revision. The authors should summarize the concerns or limitations of use of EVs when it is ready to be administrated in clinic in the last section of the manuscript. 

Author Response

This is a very detailed review for the translation of mesenchymal stromal cells derived EVs for the treatment of tendon and ligament injuries. The manuscript can be accepted after one minor revision. The authors should summarize the concerns or limitations of use of EVs when it is ready to be administrated in clinic in the last section of the manuscript.  

Reply: Thank you very much for your comments. A paragraph summarizing the limitations of using EVs is added in the last section of the revised manuscript.

There are technical challenges for clinical application of MSC-EVs. First, the biochemical composition of MSC-EVs remains unclear. The production or uptake mechanisms are poorly described. GMP standards for clinical grade production, storage and recovery are therefore lacking. Second, the drug loading efficiency of EVs is relatively lower than that for liposomes. Third, engineered MSC-EVs containing transgene products are categorized as ATPs and hence need to meet more stringent regulatory requirements. Fourth, as the half-life of MSC-EVs is short and they cannot replicate themselves like MSCs, large amount of MSC-EVs is required for clinical application. Large scale and efficient production of MSC-EVs remain difficult at present. Finally, the physiochemical properties, particle size and concentration as well as content of MSC-EVs have been the focus of investigation. The MoA or potency of MSC-EVs remains relatively unexplored, resulting in the lack of appropriate functional assays. To overcome the bottleneck for MSC-EV-based therapies, clinical-grade MSCs meeting requirements for therapeutic use and transplantation are needed for MSC-EV production. Conditions to ensure sustainable release of MSC-EVs should be established. The MSC-EV specification varies with the enrichment methods. Therefore, it is important to choose a cost-effective, scalable and automatable concentration method as early as possible to avoid changing the method and hence MSC-EV specification due to scaling up of production. While there is some consensus of using -80 oC for MSC-EV transportation and storage, storage at -80 oC poses challenges in transportation and is a cost-ineffective approach. Alternative methods such as lyophilization may improve MSC-EV stability during storage and transportation. The route of administration greatly influences the therapeutic efficacy and safety of MSC-EVs and should be determined prior to clinical application. Effort to develop high-throughput and precise quantification methods for assessing MoA or potency will greatly facilitate clinical translation of MSC-EV-based therapies.”

Round 2

Reviewer 1 Report

In this manuscript entitled “Practical Considerations for Translating Mesenchymal Stromal Cell-derived Extracellular Vesicles from Bench to Bed”, the authors summarized the Extracellular vesicles (EVs) derived from mesenchymal stromal cells (MSCs) have shown potential for the treatment of tendon and ligament injuries. I also think that the topic may be interesting to the field. The following points should be addressed for paper re-submission.

Major comments

1)     This summary is a list of sentences and tables only and is difficult for the reader to understand. It would help the readers to have a scheme for important points.

2)     1.2: It seems to be criticizing researchers of stem cell-based therapy. Each study has its advantages and disadvantages. Please change the description to one that is widely accepted by different readers.

3)     Tables need to be written more complete for the readers.

Minor comments

1)     Line 481: The words “compared with” should be “than”. There are errors or insufficient explanations through the manuscript.

2)     Table: Both “50 uL PBS” and “50 uL of PBS” are used; the authors should maintain consistency in terminology. 

Author Response

In this manuscript entitled “Practical Considerations for Translating Mesenchymal Stromal Cell-derived Extracellular Vesicles from Bench to Bed”, the authors summarized the Extracellular vesicles (EVs) derived from mesenchymal stromal cells (MSCs) have shown potential for the treatment of tendon and ligament injuries. I also think that the topic may be interesting to the field. The following points should be addressed for paper re-submission.

Major comments

  • This summary is a list of sentences and tables only and is difficult for the reader to understand. It would help the readers to have a scheme for important points.

Response: Thank you very much for your comments. A figure summarized the key issues to consider for the manufacturing of MSC-EVs for clinical trial/practice is included in the revised manuscript.

  • 2: It seems to be criticizing researchers of stem cell-based therapy. Each study has its advantages and disadvantages. Please change the description to one that is widely accepted by different readers.

Response: Thank you very much for your comments. We have no intention of criticizing researchers of stem cell-based therapy. In fact, our team did a lot of research on MSCs. We have revised the tone of the paragraph to make the content more acceptable to different readers.

  • Tables need to be written more complete for the readers.

Response: We have revised the tables to make them concise and more easily understandable by readers.

Minor comments

  • Line 481: The words “compared with” should be “than”. There are errors or insufficient explanations through the manuscript.

Response: This is corrected.

  • Table: Both “50 uL PBS” and “50 uL of PBS” are used; the authors should maintain consistency in terminology. 

Response: We have revised to ensure consistency throughout the manuscript.

Round 3

Reviewer 1 Report

I am satisfied with the revisions that have been made by authors.